

# Snow properties at the forest-tundra ecotone: predominance of water vapor fluxes even in thick moderately cold snowpacks

Georg Lackner[1,2,3,4], Florent Domine[2,3,5], Daniel F. Nadeau[1,4], Matthieu Lafaysse[6], and Marie Dumont[6]

[1]Department of Civil and Water Engineering, Université Laval, Québec, Canada
[2]Takuvik Joint International Laboratory, Université Laval (Canada) and CNRS-INSU (France), Québec, Canada
[3]Centre d'Études Nordiques, Université Laval, Québec, Canada
[4]CentrEau, Université Laval, Québec, Canada
[5]Department of Chemistry, Université Laval, Québec, Canada
[6]Univ. Grenoble Alpes, Université de Toulouse, Météo-France, CNRS, CNRM, Centre d'Études de la Neige, 38000 Grenoble, France

**Correspondence:** Georg Lackner (georg.lackner@mailbox.org)

**Abstract.**

The forest-tundra ecotone is a large circumpolar transition zone between the Arctic tundra and the boreal forest, where snow properties are spatially variable due to changing vegetation. The extent of this biome through all circumpolar regions influences the climate. In the forest-tundra ecotone near Umiujaq in northeastern Canada (56°33'N, 76°28'W), we contrast

the snow properties between two sites, TUNDRA (located in a low-shrub tundra) and FOREST (located in a boreal forest), situated less than 1 km apart. Furthermore, we evaluate the capability of the snow model Crocus, initially developed for alpine snow, to simulate the snow in this subarctic setting. Snow height and density differed considerably between the two sites. At FOREST, snow was about twice as deep as at TUNDRA. The density of snow at FOREST decreased slightly from the ground to the snow surface, in a pattern that is somewhat similar to alpine snow. The opposite was observed at TUNDRA, where the

pattern of snow density was typical of the Arctic. Crocus was not able to reproduce the density profiles at either site using its standard configuration. We therefore implemented some modifications for the density of fresh snow, the effect of vegetation on compaction and the lateral transport of snow by wind. We demonstrate that upward water vapor transport is the dominant

mechanism that shapes the density profile at TUNDRA, while a contribution of compaction due to overburden weight becomes

visible at FOREST. The adjustments that were made to Crocus partly compensate for the lack of water vapor transport in the

model, but are site-specific to some extent. Furthermore, the challenges using Crocus suggest that the general lack of water

vapor transport in the snow routines used in climate models leads to an inadequate representation of even thick and moderately

cold snowpacks, with possible major impacts on meteorological and climate projections.

## 1   Introduction

Seasonal snowpacks significantly increase surface albedo (Cohen and Rind, 1991) and soil insulation (Meredith et al., 2019)

and are thus critical to the planet's surface energy budget. In terms of spatial coverage, most seasonal snowpacks are found in

the Arctic tundra and boreal forest biomes, with clear structural differences between the two. In the tundra, snow is usually

shallow and has few distinct layers, whereas in the boreal forest, snow is deeper and has a more complex stratigraphy (Royer

et al., 2021a). The transition zone between both biomes is called the forest–tundra ecotone, and includes areas of short tundra

vegetation alongside forest patches. As snow height depends on the density and height of the vegetation, the resulting snow

cover is heterogeneous (Roy-Léveillée et al., 2014). Little is known about the snow structure in the forest–tundra ecotone.

Considering its extent throughout circumpolar regions, the considerable changes this ecotone is facing (e.g. rapid greening, see

Payette et al. (2001) and Ju and Masek (2016)), and its role in the global climate, more research is essential.

    The weather conditions to which Arctic snow is typically exposed differ considerably from conditions in the boreal forest

(Sturm et al., 1995; Royer et al., 2021a). During the cold season in the Arctic tundra, very low air temperatures occur together

with low precipitation and high wind speeds. The result is a shallow snowpack with strong vertical temperature gradients,

particularly in the fall when the ground is not yet frozen. Consequently, the dominant processes that shape the snowpack

structure are the upward transport of water vapor, driven by the high temperature gradient, and the wind-induced compaction

of the upper layers. This creates a low-density layer of depth hoar at the bottom and a hard, dense, wind-packed layer at the

top (Domine et al., 2015, 2016b). Contrarily, in the boreal forest, air temperatures and precipitation are typically higher, while

wind speeds are lower. Thus, the snowpack is thicker and the vertical temperature gradient is smaller (Royer et al., 2021a). This

is particularly true for the boreal forest of northeastern Canada, where precipitation is relatively high compared to Alaska or

Siberia (Groisman and Easterling, 1994). In forested environments, the compaction of the lower snow layers due to the weight

of the upper layers becomes the dominant process, similar to alpine snow. This results in a snow cover that is spatially complex,

where patches of typical Arctic snow blend into patches of alpine-like snow (Morin et al., 2013).

Detailed snow models like Crocus (Vionnet et al., 2012) and SNOWPACK (Bartelt and Lehning, 2002; Lehning et al.,

2002a,b) have been applied in the Arctic with limited success. Studies using both Crocus (Barrere et al. (2017) and Royer

et al. (2021b)) and SNOWPACK (Gouttevin et al., 2018) show that this type of snow is generally not well-modeled, as the

simulated density profiles do not match the observations. The lack of consideration of water vapor fluxes is suspected to be

one of the main reasons for this (Domine et al., 2019). These models have been developed for alpine applications (Brun et al.,

1989; Bartelt and Lehning, 2002), so the dominant process that controls the density profile in the models is the compaction

that results from overburden weight. To overcome this deficiency, Barrere et al. (2017), Gouttevin et al. (2018) and Royer

et al. (2021b) all introduced modifications by increasing the maximum density of wind-induced snow compaction and adapting

compaction to vegetation characteristics. This considerably improved the simulated density profiles and made them more

comparable to observations at the site scale. On the other hand, simulations in the boreal forest have so far focused on the

snow water equivalent (SWE). Studies in Canada (Bartlett et al., 2006; Oreiller et al., 2014) and Eurasia (Brun et al., 2013;

Decharme et al., 2016) showed that the bulk density and the SWE could be simulated reasonably well within the boreal forest. However, the ability of those models to adequately simulate density profiles has yet to be tested.

Considering the lack of information on the internal physical properties of subarctic snowpacks as well as on the performance of current snow models to accurately simulate these properties, we present snowpit data from two sites in the subarctic region and test the snow model Crocus at both sites. One site was located in a tundra environment and the second one was in a forest nearby. At both sites, we measured profiles of density, thermal conductivity and temperature of the snow. The snow density profiles at both sites were then compared to Crocus simulations to evaluate its performance. Furthermore, we explored adjustments to Crocus similar to those from Barrere et al. (2017) and Royer et al. (2021b) to assess their effectiveness in a subarctic setting.

## 2 Methodology

### 2.1 Study Site

Our study site was located in the Tasiapik valley, near the village of Umiujaq, Quebec, Canada (56°33'31"N, 76°28'56"W), on the eastern shore of Hudson Bay. The valley is 4.5 km long and 1.3 km wide, with elevations ranging from 0 to over 350 m above sea level, and is at the transition between the boreal forest and the Arctic tundra (Figure 1). While the upper part of the valley is dominated by lichen and shrub tundra, the vegetation in the lower part consists of a mixture of forest and high shrubs. The shrubs (mainly dwarf birch *Betula glandulosa*) are between 0.2 m and 1 m tall and cover 70 to 80% of the upper valley. The trees in the lower valley consist of black spruce (*Picea mariana*) up to 5 m tall and are estimated to cover roughly 20% of the surface, while the majority is covered by medium-height shrubs (*Salix spp.* and *Betula glandulosa*) with willows





reaching 2-3 m in height. Soils are predominantly sandy (Lemieux et al., 2020). While the soil in the upper part of the valley

consists almost exclusively of sand (>90%), the sand fraction is lower in forested areas, although no detailed measurements

were available. For more details about the study site, see Lackner et al. (2021) and Gagnon et al. (2019).

**Figure 1.** (a) Location of the study site on the eastern shore of Hudson Bay, Quebec, Canada. (b) View of the Tasiapik valley towards the south-west, where the bottom (left) is covered with trees and the top (right) with tundra. Note the presence of the cuesta that delimits the valley. (c) FOREST station. (d) TUNDRA station

## 2.2  Instrumental Setup

Two stations were deployed in the valley. One was located in the middle part (FOREST, ≈80 m above sea level, see Figure

1c) and the other was in the upper part (TUNDRA , ≈140 m above sea level, see Figure 1d), with a distance of about 900 m

between the two. The full radiation budget (CNR4, Kipp and Zonen, The Netherlands), air temperature and relative humidity

(model HMP45, Vaisala, Finland), wind speed (A100, Vector Instruments, UK), and snow height (SR50, Campbell Scientific,

USA) were measured at 2.3 m above ground at both sites. Additional measurements of wind speed and direction at a height of

10 m (model 05103, R.M. Young, USA), specific humidity (IRGASON, Campbell Scientific, USA), and precipitation (T200B,

Geonor, USA) were also collected at TUNDRA, some 20 m away. Snow temperature and thermal conductivity were measured

at both sites using vertical poles equipped with 5 TP02/TP08 heated needle probes (Hukseflux, The Netherlands). At TUNDRA,

a second pole held 18 T-type thermocouples and 4 TP08 heated needle probes. The measurement principle of the TP02/TP08

heated needles is detailed in Morin et al. (2010) and Domine et al. (2015, 2016b). In short, each needle has a heated section and

an unheated reference thermometer. The temperature of both is recorded when the needle is heated. The temperature difference

between the two parts is then plotted against the logarithm of the time elapsed since the onset of heating. The effective snow

thermal conductivity $k_{eff}$ is inversely proportional to the slope of the resulting regression line. According to Domine et al.

(2016b), the resulting error in the thermal conductivity can be as high as 29%. Lastly, three time-lapse cameras were installed

at TUNDRA.

Snow field surveys were conducted once or twice a year from 2012 to 2019 at different times throughout the winter (from

January to April). During each field survey, snow pits were dug at several locations around the two study sites and further away

in a perimeter of several hundred meters encompassing the site, with similar vegetation. A subset of the snow pit data (from

2012 to 2015) is presented in Domine et al. (2015). For each snow pit, the grain types of the layers were identified and the

density and temperature profiles were measured. For some snow pits, the thermal conductivity profiles were also measured. A

portable instrument equipped with a TP02 heated needle was used to measure thermal conductivity and a 100 cm$^3$ box cutter

(Conger and McClung, 2009) and a field scale were used to measure the density profiles.



### 2.3 ISBA-Crocus Land Surface and Snow Models

Crocus and ISBA (Interaction Sol-Biosphère-Atmosphère) are part of the SURFEX modeling platform version 8.1 (http://www.umr-cnrm.fr/surfex/) developed by Météo-France. ISBA (Noilhan and Planton, 1989; Decharme et al., 2011) simulates water and energy exchanges between the atmosphere, the vegetation, and the soil. In the presence of snow, Crocus is activated. Crocus (Vionnet et al., 2012) is a physically-based snow scheme that can distinguish up to 50 snow layers each defined by their thickness, temperature, density, liquid water content, age, and microstructural properties (optical diameter and sphericity). These properties evolve according to physical processes such as thermal conduction, snow metamorphism, and snow compaction.

A first adaption of Crocus to Arctic snow has already been introduced in Vionnet et al. (2012) in order to simulate blowing snow events. Due to the 1D nature of the model, simulating blowing snow is problematic. Two separate processes have been implemented in Crocus to simulate blowing snow, first, sublimation can be increased to simulate the loss of snow, effectively reducing the mass of the snowpack or secondly, the upper layers can be densified, without changing the mass of the snowpack. For the first option, the quantity of sublimating snow is increased due to blowing snow episodes and follows the parametrization of Gordon et al. (2006), with the corresponding mass being subtracted from the snowpack. This option was not activated in our study in order not to artificially increase sublimation rates. For the second option, enabled here, the upper snow layers are compacted according to the following equation

$$\frac{\delta \rho}{\delta t} = \frac{\rho_{max} - \rho}{\tau}.$$  (1)

where $\rho$ is the current density of the upper snow layer, $\rho_{max}$ is the maximum density (set as 350 kg m$^{-3}$) and $\tau$ is a time

scale set to 48 h. Here, we raised the maximum value $\rho_{max}$ to 600 kg m$^{-3}$, as suggested in Barrere et al. (2017) and Royer

et al. (2021b) for Arctic applications.

Barrere et al. (2017) showed that the default version of Crocus was not capable of correctly simulating density profiles that

were observed in Arctic snow. Preliminary simulations at our study site led to the same conclusion. We therefore deemed it

relevant to explore modifications to the code that are all based on physical processes specific to the Arctic environment, in

order to remedy this shortcoming. We chose to focus on three key processes that were suggested by Barrere et al. (2017),

Gouttevin et al. (2018) and Royer et al. (2021b): wind densification, particularly the densification of fresh snow (*Snowfall*),

compaction due to the weight of the snow column above (*Compaction*), and the lateral transport of snow during blowing snow

events (*Blowing snow*). Note that our goal here is not to propose an optimal parametrization of these processes, but rather to

explore whether their adaptation to the low-Arctic context can improve simulations.

For the first process *Snowfall*, there are three options for fresh snow density in the default version of Crocus (Vionnet et al.

(2012), Schmucki et al. (2014) and Anderson (1976)), as detailed in Lafaysse et al. (2017). All three depend on wind speed and

air temperature (except for the one from Anderson (1976) which depends on temperature only) and lead to rather low densities

given the cold temperatures typically found in the Arctic. As in Royer et al. (2021b), we used the parametrization of Vionnet

et al. (2012) for the fresh snow density $\rho_n$

$$\rho_n = max(50, a_\rho + b_\rho(T_a - T_{fus}) + c_\rho W s^{1/2}), \tag{2}$$



where $a_\rho$ = 109 kg m$^{-3}$, $b_\rho$ = 6 kg m$^{-3}$ K$^{-1}$, and $c_\rho$ = 26 kg m$^{-7/2}$ s$^{-1/2}$ are parameters and $T_{fus}$ is the melting point

of water. Equation 2 is driven by air temperature ($T_a$) and wind speed ($Ws$). Motivated by the fact that the top layers of

the snowpack are usually very hard in Arctic environments, we opted to increase the density of fresh snow by doubling the

parameter $a_\rho$ and multiplying $c_\rho$ by 5. These values were obtained with a sensitivity analysis where the density of the upper

snow layers served as validation data. Vegetation traps snow and prevents the subsequent transport of snow by wind (Essery

and Pomeroy, 2004), thus reducing the effect of wind on the density of fresh snow. We therefore chose to apply this new

parametrization only when the height of the snowpack exceeds the vegetation height. When that is not the case, the default

parameters from Vionnet et al. (2012) are used.

The process *Compaction* makes snow densification dependent on canopy height. This takes into account the stabilizing effect

from vegetation and follows observations of Domine et al. (2016a) that density within shrubs is significantly lower than above.

In Crocus, snow density $D$ is updated at each time step $dt$ to account for compaction. Herein, we introduce a parameter acting

as a factor $c$ of this increase in density.

$$\frac{dD}{D} = c\frac{-\sigma}{\eta}dt, \tag{3}$$

where $\sigma$ is the overburden and $\eta$ is the viscosity. This allows us to modulate the increase in density due to compaction from

0% to 100%, where 0% means no increase in density and 100% means that the default procedure is applied. In this study,

we selected a fixed value of 0.05, meaning the increase was reduced to 5% of its default value. This value was obtained by

comparing observed and simulated density profiles. Following observations from Ménard et al. (2014) and Belke-Brea et al.

(2020) highlighting the bending of shrubs under the weight of snow, this reduction in densification due to compaction is applied

only for half the height of the canopy.

Lastly, a simple scheme to compensate for the lack of a blowing snow scheme (*Blowing snow*) was implemented in Crocus.

High winds are very frequent in the Arctic, particularly during snowfall events, and thus blowing snow is extremely important

(Li and Pomeroy, 1997). Due to the 1D nature of the model, it is not able to explicitly take this phenomenon into account.

However, ignoring the effects of blowing snow would greatly alter the simulation results. As such, we implemented a linear

equation that can modulate actual precipitation during blowing snow events, to account for lateral transport. For offline simula-

tions, as presented here, no implementation of snow erosion or accumulation is available in Crocus. However, as stated earlier,

a blowing snow process is already included in Crocus, but there snow is removed by increasing sublimation. We opted for a

process that can add or remove snow without changing the sublimation in order to avoid the artificial alteration of this flux, the

precipitation rate PR was changed based on wind speed:

$$P_{new} = P_{old}(a + b\,Ws). \tag{4}$$

In equation 4, $P_{new}$ is the new precipitation rate, $P_{old}$ is the old rate, $Ws$ is the wind speed in m s$^{-1}$ and $a$ and $b$ are

coefficients. We obtained a reasonable agreement between the simulations and observations of the vertical density profiles and

snow heights with $a = 0.1$ and $b = 0.3$. To account for the fact that blowing snow does not occur at low wind speeds, this option

was only activated for wind speeds greater than 3 m s$^{-1}$. Additionally, for wind speeds higher than 10 m s$^{-1}$, the increase in

precipitation was limited to twice the original precipitation rate. Areas with tall vegetation act as sinks for wind-blown snow

(Myers-Smith and Hik, 2013). A preliminary series of tests revealed that it was desirable to use equation 4 for FOREST only, in order to remain as close as possible to the observed snow heights and focus our analysis on the internal properties of the

165 snow cover. Note that equation 4 could also be used to remove precipitation when needed, for instance when simulating highly wind-prone sites.

In summary, we explored various (non-optimized) modifications that target processes known to be poorly managed by Crocus in the Arctic. At TUNDRA, the *Snowfall* and *Compaction* modifications were enabled, while at FOREST, the *Snowfall*, *Compaction*, and *Blowing snow* modifications were all activated.

**2.4 Forcing Data**

The meteorological forcing variables of ISBA-Crocus are typical of a land surface model: air temperature, specific humidity, wind speed, incoming shortwave and longwave radiation, atmospheric pressure, and (solid and liquid) precipitation rates. Observations of these variables at each of the two sites have been collected since 2012, except for atmospheric pressure, which was available since June 2017. ERA5 (Hersbach et al., 2020) data were used for the pressure before 2017 and were adjusted

applying a simple regression obtained with data from after 2017 between the measured data and the ERA5 data ($R^2$ = 0.99 after the adjustment).

The modeled soil columns had a total thickness of 12 m and were divided into 20 layers of increasing depth. Following the soil water content analysis from Lackner et al. (2021), we also adjusted two soil hydraulic parameters, the saturated soil water content and the field capacity. The soil composition was set to 95% sand and 5% silt (Gagnon et al., 2019) for TUNDRA and to

180 80% sand, 15% silt and 5% clay for FOREST based on estimates. To ensure the equilibrium of soil moisture and temperature, we initialized the model with a spin-up of five years (2012-2017). Note that because observations of precipitation from before





| | $T_a$ ($^{\circ}$C) | | $Ws$ (m s$^{-1}$) | | $SW_{down}$ (W m$^{-2}$) | |
| --- | --- | --- | --- | --- | --- | --- |
| | mean | MAD | mean | MAD | mean | MAD |
| 2017-18 | 0.5 | 1.1 | 0.8 | 1.3 | 14.3 | 14.4 |
| 2018-19 | 0.5 | 1.1 | 1.0 | 1.5 | 8.3 | 9.8 |
| 2019-20 | 0.3 | 0.7 | 0.9 | 1.2 | 4.5$^{\star}$ | 4.8$^{\star}$ |
| all winters | 0.4 | 1.0 | 0.9 | 1.3 | 9.0 | 9.6 |

**Table 1.** Mean difference and mean absolute difference MAD of the air temperature $T_a$, wind speed $Ws$ and downwelling shortwave radiation $SW_{down}$ for three winters 2017-18, 2018-19, and 2019-20. $^{\star}$Note that the radiation at TUNDRA was replaced with FOREST data for parts of winter 2019-20 (15 December to 1 March) due to problems with the instrument at TUNDRA.

2015 were not available, raw ERA5 data had to be used for the 2012–2015 period. We corrected the precipitation data for

undercatch using the transfer function of Kochendorfer et al. (2017). For the partitioning of precipitation into rain and snow,

we used a fixed threshold of $0.5^{\circ}$C. The suitability of the threshold was tested using air temperature and observations of the

type of precipitation from Environment and Climate Change Canada (https://climate.weather.gc.ca, last access: 15 December

2021) at the Umiujaq airport ($\approx$3km away from the TUNDRA).

Thus, we used a different forcing data set for each station. Because some of the required variables were not available at

FOREST, we used the precipitation, pressure and specific humidity from TUNDRA. Given the proximity of the two sites, the

differences between these variables are presumed to be very small.

**3   Results**

**3.1   Observed Meteorological Conditions**

Before analyzing the snow characteristics, we compared the meteorological conditions at both sites (Figure 2).

Additionally, in Table 1, the mean difference and the mean absolute difference are shown for winters 2017-18, 2018-19, and

2019-20.





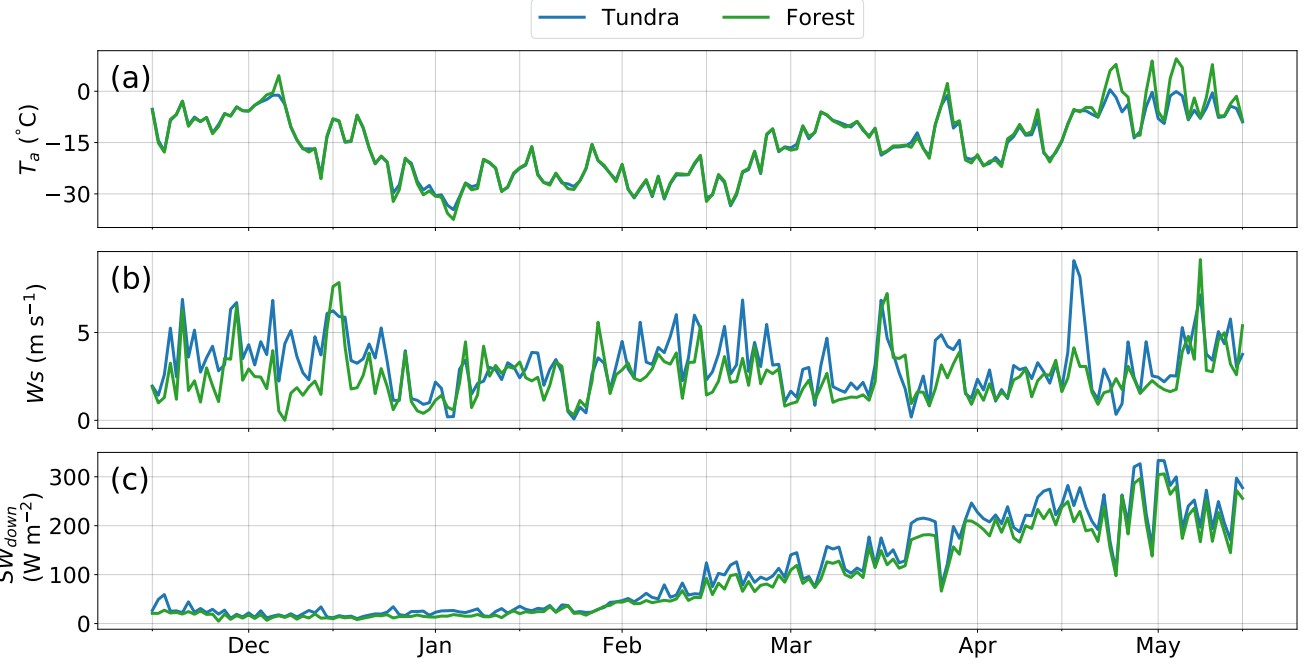

**Figure 2.** Daily mean (a) air temperature $T_a$, (b) wind speed $Ws$ and (c) downwelling shortwave radiation $SW_{down}$ measured at the two sites TUNDRA and FOREST for winter 2018-19.

Air temperatures at the two stations were fairly similar. On average, TUNDRA was $0.4^\circ$C colder than FOREST. Only

occasional minor differences ($\lesssim 2^\circ$C) occurred until mid-April. After this point, air temperatures at TUNDRA were slightly

colder than at FOREST. This trend was observed for all winters, with the exception of winter 2016–17, where the difference

was less pronounced. Wind speeds were consistently higher at TUNDRA, with a mean difference of about 0.9 m s$^{-1}$ for all

years. Lastly, the disparity in downwelling shortwave radiation is also minimal, with a mean difference of 9.0 W m$^{-2}$. The

absolute difference in downwelling shortwave radiation between TUNDRA and FOREST increased in spring from February

onward, while the relative difference compared to the magnitude of the fluxes remained comparable throughout winter. The

difference likely arises due to the location of FOREST further down the valley, where topographic shading is more significant.





The cumulative solid precipitation for three years is shown in Figure 3. Despite fairly variable interannual amplitudes, the temporal patterns were very similar from one year to the next. In fall, snowfall rates were quite sustained until mid-January,

decreased temporarily and then rose again in April.

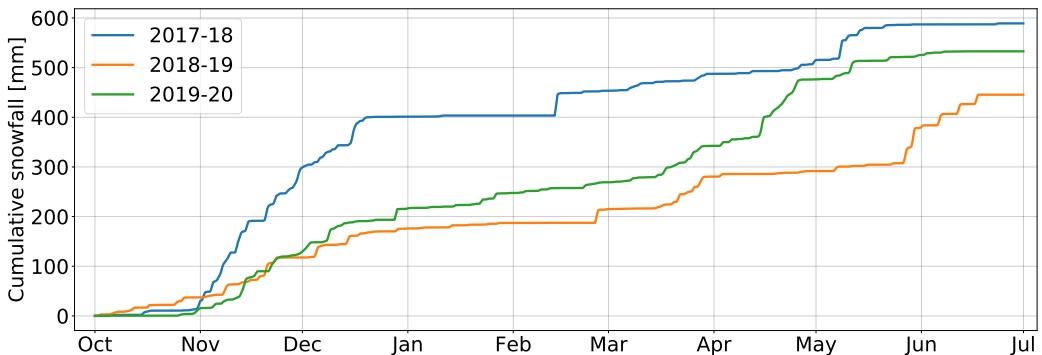

**Figure 3.** Cumulative snowfall for three winters.

## 3.2 Observations of Snow Properties

### 3.2.1 Snow Height

Figure 4 shows the evolution of observed snow height for five consecutive winters (2015–2020) at TUNDRA and for three winters at FOREST.

The onset of a permanent snow cover consistently occurred in the second half of October each year. Also, the snow cover formed at the same time at both sites. However, the subsequent evolution exhibited some differences between the two sites. At TUNDRA, a large fraction of the snow accumulation took place in the first few months of winter, up to mid-January. This was usually followed by a period of low precipitation (see Figure 3) and thus low snow accumulation. Towards the end of winter, an increase in snowfall coincided with peak snow heights, typically in April or early May. At FOREST, the accumulation was

more evenly distributed over the entire winter and a gradual increase in snow height until April or early May was observed for





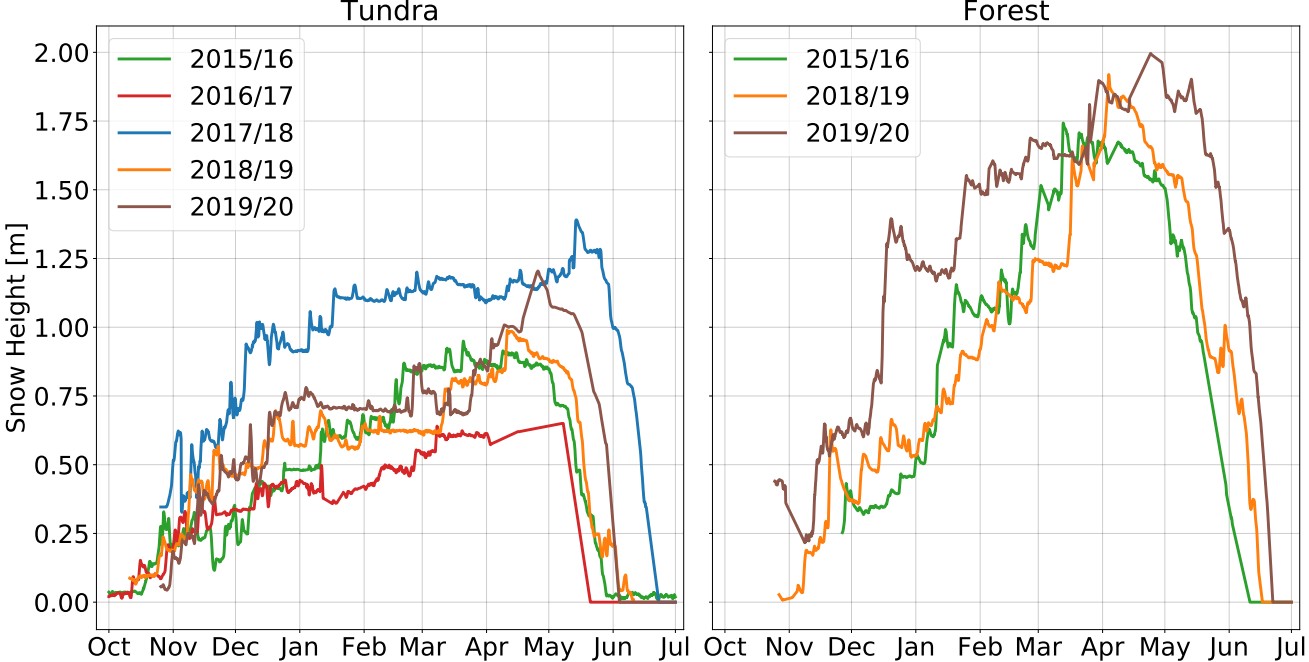

**Figure 4.** Comparison of the measured snow height of TUNDRA and FOREST for several years. When necessary, missing data were filled in with time-lapse camera observations of a graduated rod. Unfortunately, there was too much missing data from the winters 2016-17 and 2017-18 at FOREST, so those winters are not presented here.

all years. The melt-out date at TUNDRA was strongly correlated with maximum snow height, and on average, this occurred in early June. However, depending on the maximum snow height, the melt-out date could occur half a month earlier or later, as was observed in winters 2016–17 and 2019–20. As the maximum snow height at FOREST was much more similar between the years, the melt-out dates were also more consistent and took place around mid-June for all years.

Although the observations presented in Figure 4 are a good proxy for determining general snow heights at the sites, there is a high spatial variability. This variability is caused by the redistribution of snow by frequent high winds combined with differences in micro-topography and vegetation. For instance, on 12 April 2018, we sampled the snow height within a 100 m radius of TUNDRA and observed large variations in snow height. The heights varied between 50 cm and 210 cm, with a mean value of 109 cm. This is within 8 cm of the height measured by the automatic station that day (117 cm).





### 3.2.2 Stratigraphy


Differences in snow heights are reflected in the internal composition of the snow cover. Figure 5 shows simplified stratigraphies

of representative samples from both TUNDRA and FOREST.

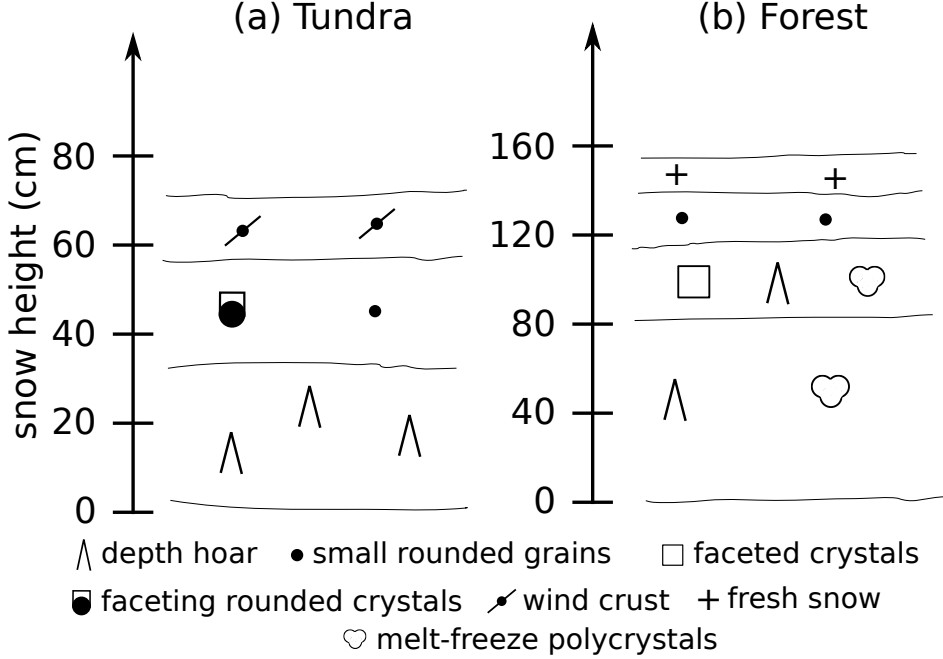

**Figure 5.** Simplified stratigraphies from February 2014 representing typical snow conditions at sites (a) TUNDRA (25 February) and (b) FOREST (24 February). Note that the y-scales are different for the two sites.

At TUNDRA, the depth hoar made up around half of the total depth. Just above the depth hoar, some layers of faceting or

rounded faceted crystals were present, whereas the top of the snowpack usually consisted of a hard wind slab. The fraction of

each layer type was highly variable. Furthermore, there were often more than three layers observed, and wind slabs alternating

with layers of faceted crystals were fairly common.

The stratigraphy at FOREST was markedly different from that found at TUNDRA. While the depth hoar fraction was

comparable to the one found at TUNDRA, melt-freeze forms were often present within these basal layers. On top of the depth





hoar layers, there were often layers of small, rounded crystals. While the uppermost layers at TUNDRA were usually made up

of a wind slab, fresh snow (precipitation particles) was often found in the top layer of the snowpack at FOREST, as seen in

Figure 5.

### 3.2.3   Density and Thermal Conductivity

On the left side of Figure 6, the density profiles of 29 snow pits that were dug in the months of January, February, and March

from the years 2012–2019 are shown, along with their means. They were all dug in the surrounding area of TUNDRA or

similar environments. The same is shown on the right side of Figure 6, but for 18 snow pits that were dug near FOREST or in

similar environments. In order to make the profiles comparable, the snow heights were normalized.

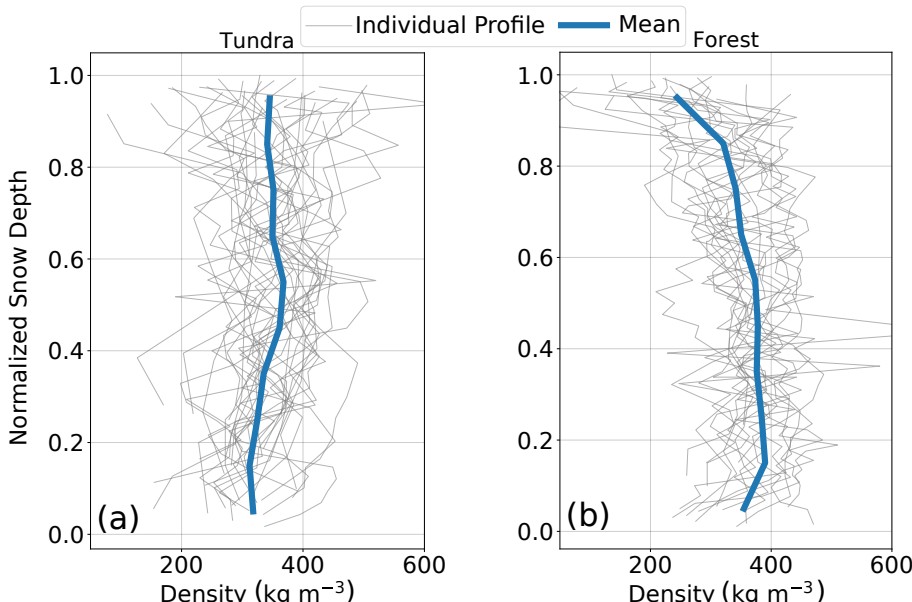

**Figure 6.** Snow density profiles from 29 snow pits in tundra environments and 18 snow pits in forested environments collected between January and March from the years 2012 to 2019. For better comparability, snow heights were normalized. The means of all profiles are also shown.





Due to the contrast in snow heights between the two sites, the vertical density profiles also showed significant differences.

While mean snow density slightly increased with height at TUNDRA, there was a clear decrease in density for the upper 80%

of the snowpack at FOREST (only the lowermost snow layer did not follow this general trend). At TUNDRA, the mean density

at the bottom of the snowpack was around 315 kg m$^{-3}$ and then rose to 350 kg m$^{-3}$ in the middle and upper parts of the

snowpack. At FOREST, snow in the basal layers had a mean density of around 375 kg m$^{-3}$, which decreased to less than 250

kg m$^{-3}$ at the top. The scatter in the measured profiles was comparable at both sites, with extremes of 100 kg m$^{-3}$ on either

side of the mean.

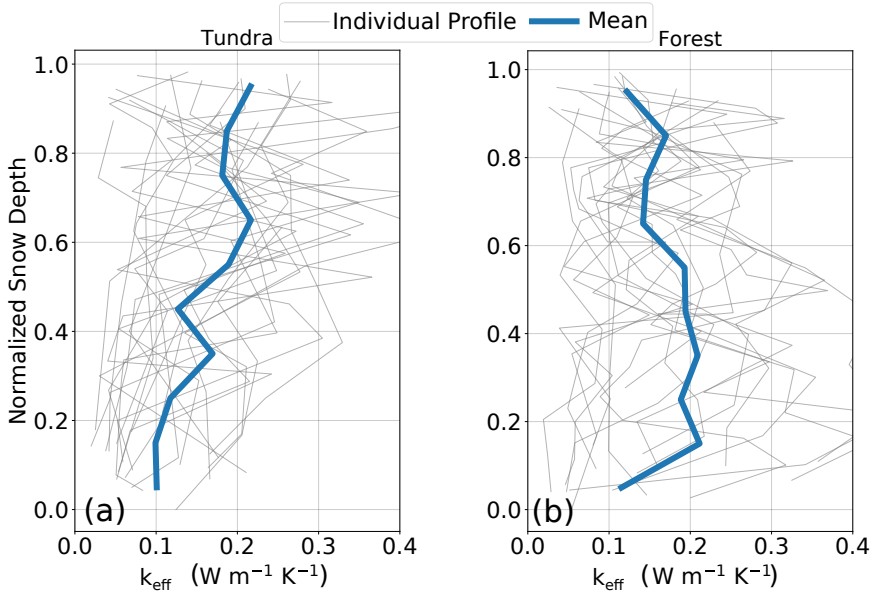

**Figure 7.** Same as Figure 6 but for snow thermal conductivity. Note that only data from 21 snow pits for TUNDRA and 17 for FOREST were used, as thermal conductivity measurements were not collected for every snow pit

The profiles of snow thermal conductivity in Figure 7 follow the same trend as the snow density in Figure 6, with $k_{eff}$

increasing with height at TUNDRA and decreasing with height at FOREST. At TUNDRA, $k_{eff}$ values increased from 0.1 W

m$^{-1}$ K$^{-1}$ in the depth hoar layers to slightly more than 0.2 W m$^{-1}$ K$^{-1}$ in the wind slab. At FOREST, the thermal conductivity





generally decreased from 0.2 W m$^{-1}$ K$^{-1}$ in the basal layers to almost 0.1 W m$^{-1}$ K$^{-1}$ in the top layer. However, the lowermost

snow layer at FOREST did exhibit very low thermal conductivity ($\approx$0.1 W m$^{-1}$ K$^{-1}$), indicating a departure from the general

trend of the profile. The scatter of the measured thermal conductivity profiles is more pronounced at both sites when compared

to that observed for density profiles, with a 178% increase of the variance at TUNDRA and 214% at FOREST. This scatter is

particularly large in the layer with the highest thermal conductivity (e.g. the top layer at TUNDRA and the near-bottom layer

at FOREST), with values ranging from 0.05 W m$^{-1}$ K$^{-1}$ to over 0.4 W m$^{-1}$ K$^{-1}$.

### 3.2.4   Soil and Snow Temperatures

Snow heights and thermal conductivity were quite different from one site to another and consequently, the same variation can

be assumed for snow and ground temperature (Figure 8).

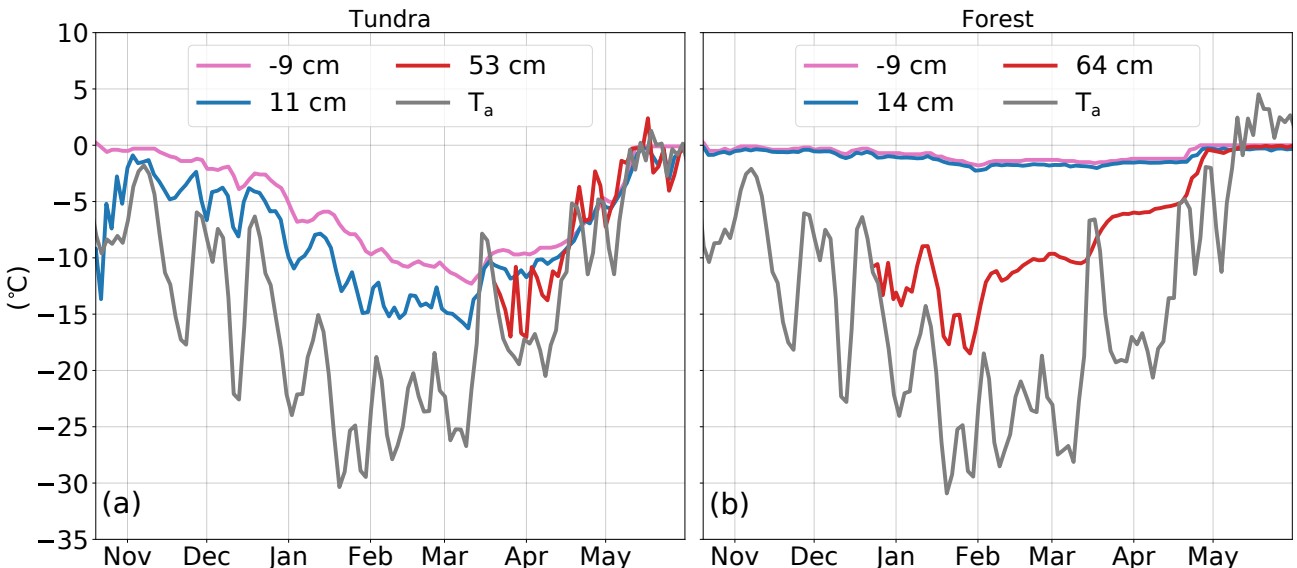

**Figure 8.** Snow (blue and red), ground (pink curve) and air temperatures (grey curve) at the (a) TUNDRA and (b) FOREST for winter 2018-19. Heights at which measurements were taken are relative to the surface of the ground. Air temperature T$_a$ was measured at 2.3 m above ground. The values correspond to measurements collected once every two days.

As air temperatures and other meteorological forcings (Figure 2) only slightly varied between the two sites, differences

between snow and ground temperatures likely arose from discrepancies in the snowpack properties. The most noticeable

divergence occurred near the soil-snow interface. At FOREST, the ground remained unfrozen until January, and then dropped

to its lowest value at slightly below $-1^\circ$C. The snow temperature at 14 cm very closely followed the ground temperature, but

was $1^\circ$C colder. At TUNDRA, the ground at 9 cm depth froze as early as mid-November and then in March dropped to the

lowest values of below $-11^\circ$C, about $10^\circ$C colder than at FOREST. At TUNDRA, the snow temperature at 11 cm was on

average $3^\circ$C colder than the ground temperature. The temperatures higher up in the snowpack (53 cm for TUNDRA and 64 cm

for FOREST) followed air temperature fluctuations more closely. At TUNDRA, the difference between air and top layer snow

temperatures varied between 0 and $5^\circ$C. At FOREST, this difference reached up to $10^\circ$C, with an evident decoupling between

air and top layer temperatures beginning in early February.

## 3.3   Modeling

### 3.3.1   Snow Height

In Figure 9, the snow heights at TUNDRA and FOREST during winters 2018–19 and 2019–20 are compared to two different

simulations, one using the default configuration of Crocus and one using the adjusted version of Crocus presented in section

2.3.

At TUNDRA, the default version shows reasonable agreement with the observations. In winter 2018–19, snow height is

underestimated by 15 to 20 cm during the accumulation period. In late April and May, the simulated melting occurs too early,

leading to a snow disappearance date that is 12 days earlier than the observations. For winter 2019–20, there is better agreement

between the default version of Crocus and the observations with a mean negative bias of 10 cm, leading to a modeled melt-out



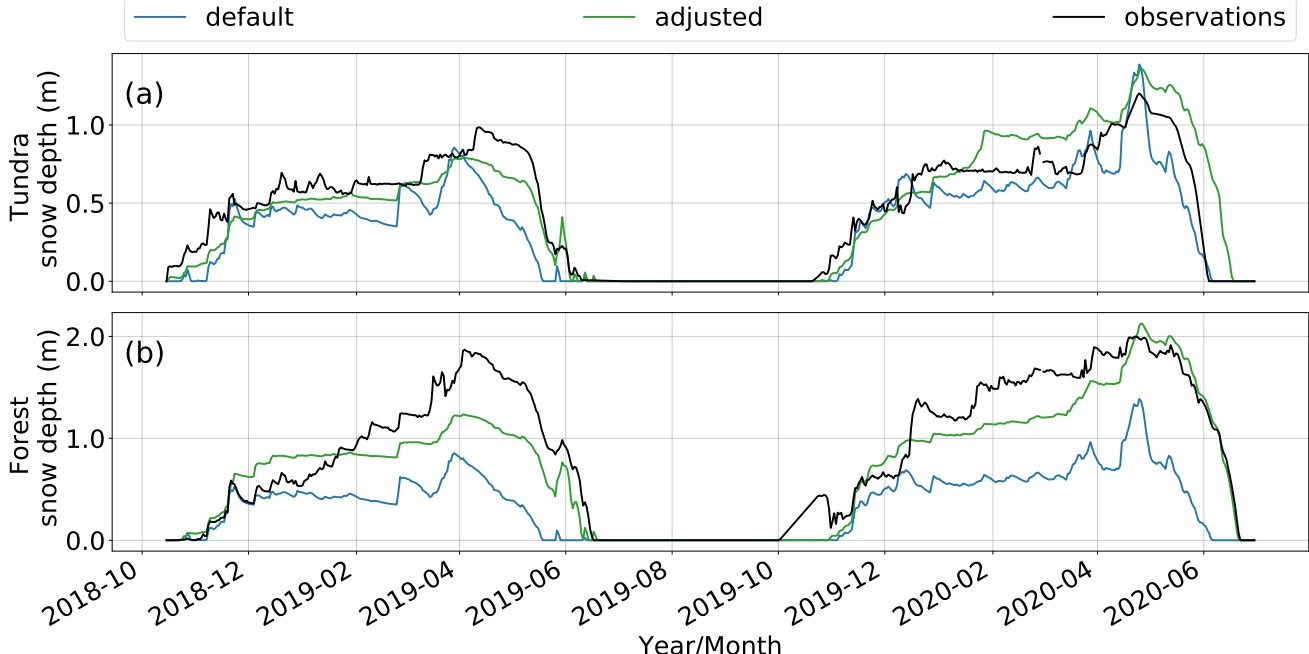

**Figure 9.** Evolution of simulated and observed snow heights during winters 2018-19 and 2019-20 (a) TUNDRA and (b) FOREST. Two different simulations are shown: the default version of Crocus and the adjusted version, in which modifications to the compaction and snowfall modules are activated at the two sites. At FOREST, a blowing snow module is implemented to account for snow transport by the wind.

date that is just two days later than the observed date. There is one exception during a precipitation event in late April in which

the snow height is clearly overestimated by the model. Simulations with the adjusted version of Crocus for TUNDRA show an

increased snow height of 10 to 20 cm compared to the default version. For winter 2019–20, this leads to a delayed melt-out that

is 15 days later than the observed date. One striking difference between the two versions is that the snow height fluctuations

are dampened in the adjusted version of Crocus. Particularly, there is much less compaction following precipitation events and

it is more in line with the observed snow height pattern.

Since the meteorological forcing is nearly the same at both sites, it is not surprising that the snow heights modeled by the

default version of Crocus at FOREST are very similar to those at TUNDRA. As a result, the modeled snow heights of the

default version are lower than the observed snow heights by a factor of 2. Thus, the simulated melt-out date is early by one

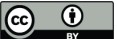

month for winter 2018–19 and by 16 days for winter 2019–20. The adjusted version of Crocus simulated the snow heights that

are much closer to those observed. However, snow heights are still underestimated by between 10 cm and 70 cm. The melt-out

date is better simulated, with a difference of 9 days in winter 2018–19 and 0 days in 2019–20.

One striking feature of all versions of the model is that simulated snow accumulation events do not always match with the

observed accumulation events. This is most noticeable in February and March, 2019. In February, all simulations show an

accumulation, while no change in snow height was observed at TUNDRA. The opposite happened in March, when an increase

in snow height was observed at the site but no accumulation is reported in the simulation. This mismatch between observations

and simulation is due to the transport of snow by wind.

### 3.3.2   Density

The mean observed density profiles (Figure 6) are compared to the default and adjusted model runs in Figure 10.

Again, the default version of Crocus produces practically the same results at both sites, considering the small differences

between the two forcing files. The mean from the default version shows a steep decline in density with height, as is typically

observed for alpine snow. At the bottom of the snowpack, the density reaches almost 500 kg m$^{-3}$, whereas the snow is very

light at the top, with a minimum density of below 90 kg m$^{-3}$. At TUNDRA, the adjusted model fairly accurately simulated the

density profile. It overestimates the density by $\approx$50 kg m$^{-3}$ in the bottom 40% of the snowpack, while it underestimates the

density by 40 to 70 kg m$^{-3}$ in the upper 60%. However, whereas the mean absolute error of the default version is 127 kg m$^{-3}$,

it declines to 38 kg m$^{-3}$ for the adjusted version. Moreover, the residual between the observations and the adjusted version is

smaller than the variance of the observations.





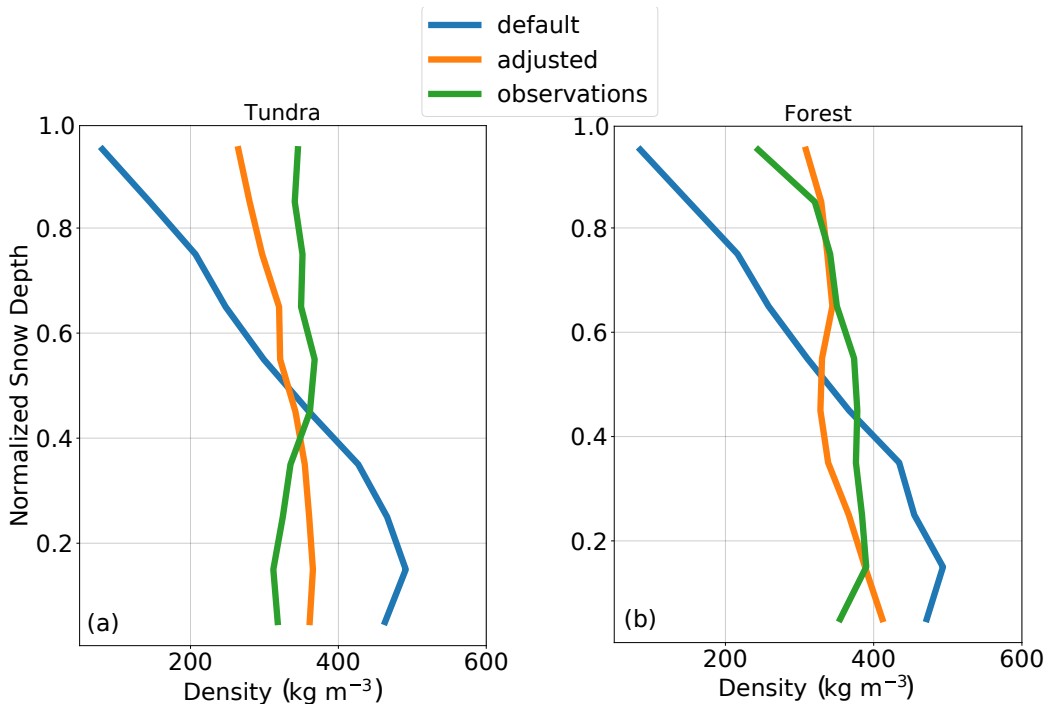

**Figure 10.** Comparison between observed mean snow density profiles from Figure 6 and the simulated density profiles from the default and adjusted versions of Crocus for (a) TUNDRA and (b) FOREST. The modeled mean profiles were obtained by averaging one snow profile per week between the months of January and March for both sites from 2017 to 2020.

At FOREST, the default version of Crocus performs better than at TUNDRA, as observations showed a decrease in density with height. However, the density at the bottom of the snowpack is still largely overestimated ($\approx$100 kg m$^{-3}$) and even more so at the top ($\approx$160 kg m$^{-3}$). Again, the adjusted version does reproduce the density profile significantly better than the default version. The mean absolute error decreases from 87 kg m$^{-3}$ for the default version to 30 kg m$^{-3}$ for the adjusted version. Similar to the profiles at TUNDRA, the variance of the observations is higher than the residual between the adjusted model and observation profile.

## 4 Discussion

### 4.1 Observations

We contrasted snow conditions at two sites located less than 1 km apart in the forest–tundra ecotone, each with very different

vegetation characteristics (shrub tundra versus open forest). Meteorological conditions differed very slightly between the two

sites (Figure 2). At TUNDRA, air temperatures were a bit colder in spring for some of the years we studied and incoming

shortwave radiation was slightly higher than at FOREST. The impact of air temperature differences on snow cover was modest,

as the largest deviations between the two sites were minor and confined to short periods. The difference in downwelling

shortwave radiation can be explained by the location of FOREST further down the valley, as for low solar angles in mid-winter,

topographic shading is more pronounced there. Again, this does not heavily impact the snowpack, as the albedo during this

period is very high ($\approx$0.85, see Lackner et al. (2022)). The high albedo means that most of the additional radiation at TUNDRA

is reflected. The higher wind speeds at TUNDRA, however, have a considerable influence on the snowpack, leading to greater

compaction and more fragmented crystals than at FOREST. The difference in wind speed between the two sites is most likely

due to the lower surface roughness at TUNDRA, resulting from the shorter vegetation.

In addition to altered snow compaction at the surface, snow is also transported by the wind. During periods of high wind

speeds ( $>3$ m s$^{-1}$), snow deposited on the surface is lifted up in the air and is typically transported several hundred meters

away before settling back on the surface (Takeuchi, 1980). This phenomenon explains the large difference in snow height

between the two sites, despite the likely similar amounts of total precipitation. As blowing snow events occur several times per

week in the valley, snow is continuously transported from the upper parts of the valley (TUNDRA) to the lower part (FOREST).

This leads to a gradual increase in snow height at FOREST. We see evidence that the taller vegetation at FOREST traps snow
early in the season. Snow surveys that have been conducted at the very top of the valley revealed a very thin snowpack ($\lesssim 40$ cm). The snow height at TUNDRA is more closely correlated with the precipitation rate, which is typically lower from January to March (see Figure 3). As a result, the maximum snow height is on average twice as high at FOREST than at TUNDRA.

The differences in stratigraphy and density are a direct result of the different snow heights at the two sites. At FOREST, where there is more snow, the bottom layers are more compacted by the weight of the overburden snow and are consequently denser. The thick snow cover insulates the soil more efficiently, maintaining warmer soil and creating a greater vertical gradient of snow temperatures throughout the winter (see Figure 11). These conditions favor the kinetic growth of snow crystals and the development of thick depth hoar layers (Marbouty, 1980). At TUNDRA, where the snow height is considerably lower, the

overburden weight and therefore the compaction of the lower layers are reduced, leading to basal layers of lower density. Due to the reduced snow height and fairly similar snow thermal conductivity, soil freezes earlier in the winter. Thus, the vertical gradient of snow temperature is greater at TUNDRA than at FOREST in early winter. However, the gradient decreases when soil freezes (Figure 11), resulting in a comparable depth hoar fraction as at FOREST. At the top of the snowpack, wind speed shapes both the density and the stratigraphy. The lower wind speeds at FOREST lead to snow that is not as densely compacted

at the top snow layers, which tends to preserve the shape of the snow crystals on the surface. This explains the observed lower density and higher abundance of precipitation particles. The opposite is true for TUNDRA, where wind speeds are higher, leading to denser top layers and precipitation particles that are rapidly fragmented and compacted into wind slabs.

Several studies have proposed a relationship between snow density and its thermal conductivity, often via a second-order polynomial (e.g. Sturm et al. (1997), Calonne et al. (2011), and Fourteau et al. (2021)). However, a key factor that is not

included in these equations is the snow type. Here, we hypothesize that this is the reason for the mismatch beetween the observed profiles of thermal conductivity in Figure 7 and those of density in Figure 6. The thermal conductivity profile at

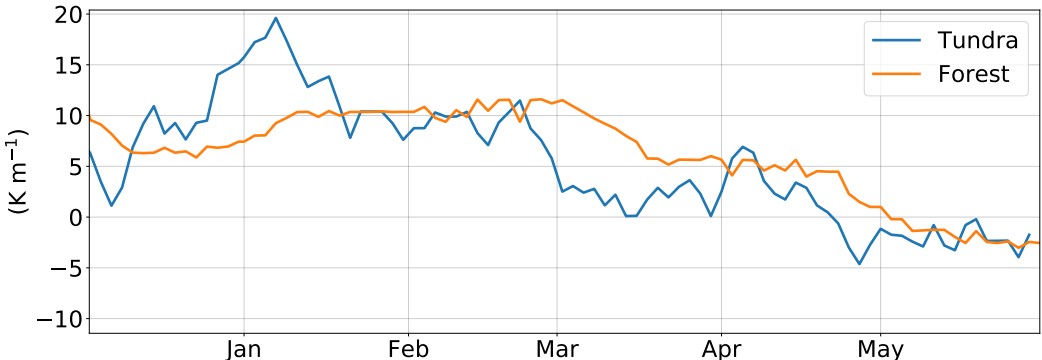

**Figure 11.** Temperature gradient at the base of the snowpack for winter 2017-18 at TUNDRA and FOREST. At TUNDRA, measurements were made at 11 cm and 21 cm while the measurement heights were 4 cm and 14 cm at FOREST. Note that the measurement interval was two days, thus the peak gradients are likely much higher than shown here.

TUNDRA shows a very clear trend of increasing values with height, while there is only a very modest increase in density with height. One should note however, that this could also be an artefact of the manual needle probe method, as the soft snow at the bottom of the snowpack often breaks when the needles are inserted. For FOREST, the decreasing trend of thermal conductivity with height is more consistent with the density profile.

## 4.2 Simulations

The accurate simulation of Arctic snow is complex, as environmental factors are strikingly different compared to those in alpine environments, for which most of the sophisticated snow models have been developed (e.g. Brun et al. (1989) and Bartelt and Lehning (2002)). In recent studies, Arctic snow was simulated using Crocus and SNOWPACK snow models that included adaptations to account for higher wind compaction and the stabilizing effect of vegetation (Barrere et al., 2017; Gouttevin et al., 2018; Royer et al., 2021b). With these modifications, the simulations of the density profiles were significantly improved and thus more realistic. The changes proposed in those studies are similar to those we have implemented in our study, but some differences should be highlighted. Barrere et al. (2017) increased only the maximum density related to the parametrization of

wind-induced snow compaction. Gouttevin et al. (2018) and Royer et al. (2021b) reduced the snow density in the vegetation,

similar to what we have done in our study. However, we also tried to account for the bending of shrubs and as such, snow

compaction was reduced for only 50% of canopy height, and not the full height. The latter two studies also implemented

different parametrizations for the density of fresh snow. While Gouttevin et al. (2018) based their parametrization on Antarctic

data from Groot Zwaaftink et al. (2013), Royer et al. (2021b) used different parameters in the formulation of Vionnet et al.

(2012) than those used here, namely they doubled the parameter $c_\rho$. Therefore, while there are clear similarities between the

various approaches, the exact parametrizations used differ, which raises the question of the degree to which the parametrizations

can be valid beyond their site of optimization.

The stabilizing influence of vegetation on the snow cover is not well documented. Ménard et al. (2014) observed and modeled

the bending of shrubs, showing that they do not just remain at the same height when being buried in snow, but rather bend

to some extent. This suggests that the snow cover is not stabilized over the full height of the vegetation, but rather on some

fraction of it. Belke-Brea et al. (2020) investigated allometric equations for the exposed vegetation fraction of shrubs at the

same site as in this study and explained deficiencies of the equation by a twofold structure of the shrubs, where the lower part

is more stable while the upper part is bent stronger. Thus, not taking the whole vegetation height as a zone where compaction

is reduced seems a reasonable choice, particularly for higher shrubs as at FOREST.

Sustained high wind speeds are common in the Arctic and are thus an important factor influencing the snowpack. First,

top snow layers are compacted by wind and including this process into the model significantly improved the results. The

second impact of strong winds is the transport of snow. This process is getting increased attention for snow simulations in

mountainous environments (Marsh et al., 2020; Vionnet et al., 2021). We argue that it is equally important for applications in

the Arctic environment. The simulated snow heights using the default version of Crocus differed from the observations by more

than a factor of 2. Thus, future studies that use distributed snow models should take snow transport into account. Combined
with the impact of the snow height on the soil temperature (see section 3.2.4), the importance of wind-driven snow transport is
apparent and has a crucial impact on permafrost studies.

Finally, we would like to emphasize that we did not set out to find the best set of coefficients to simulate the two study
sites. By including some key processes that are specific to the Arctic, our aim was to demonstrate that it is possible to simulate
vertical density profiles reasonably well. However, as the model did not perform as well at TUNDRA as it did at FOREST, our
results suggest that there is no single set of parameters that is best suited for both environments. The reason may be that these
coefficient-fitting schemes are just compensating for the lack of description of upward water vapor fluxes in snowpacks, and
the solution may be to actually include them in models.

## 4.3  Water Vapor Fluxes

At TUNDRA, the snowpack showed a significant fraction of depth hoar, a snow type that solely originates under conditions
with water vapor transport. Furthermore, the snow density increased towards the snow surface, whereas the opposite, a strongly
decreasing pattern was simulated by Crocus, where compaction by overburden weight is the most important factor influencing
the density. As our site receives a relatively large amount of precipitation, the findings show that water vapor transport is
dominant over compaction even for thicker (up to 1 m) Arctic snowpacks. Domine et al. (2016b) and Domine et al. (2019),
demonstrated this predominance for thinner Arctic snowpacks and thus, water vapor transport can be considered prevalent in
all Arctic regions.

At FOREST, we found a depth hoar fraction similar to that at TUNDRA. While the density slightly decreases towards the

surface, its gradient is still far inferior compared to simulations by Crocus. This leads to the conclusion that water vapor fluxes

also play a significant role for thick, moderately cold snowpacks, present at FOREST.

Together with Sturm and Benson (1997), who demonstrated the importance of depth hoar for thinner ($<1$ m) snowpacks

in forested subarctic regions, the results show that vapor fluxes have a critical impact on snowpack physics in the Arctic, the

(thin snowpack) boreal forest and the forest-tundra ecotone with thick snowpacks, and therefore presumably in the boreal

forest with thick snowpacks as well. Consequently, vapor fluxes cannot be neglected on the overwhelming majority of seasonal

snowpacks.

## 5   Conclusions

We analyzed several years of snow survey data at two sites that were located less than 1 km apart in the forest–tundra ecotone.

One site was covered with sparse forest (FOREST) and one was an Arctic tundra (TUNDRA). We compared the snow properties

of the two sites in terms of snow height, stratigraphy, density, thermal conductivity and snow temperature. Additionally, we ran

simulations with the snow model Crocus to explore model performance in both environments.

All the observed snow properties revealed marked differences between the two sites. Snow height was up to twice as high at

FOREST than at TUNDRA, due to wind-induced snow transport. This strongly affected the temperatures at the bottom of the

snowpack, as they were at times more than $10^{\circ}$C colder at TUNDRA than at FOREST. A large difference was also observed for

the vertical density profiles. Snow density slightly increased with snow height at TUNDRA, indicating a profile that is typical

of Arctic snow. In contrast, the density at FOREST decreased with height, which is more typical for alpine environments. In

both cases, a substantial depth hoar layer occupied the lower portion of the snowpack.

The Crocus model failed to reproduce the density profile of both sites. Non-optimized adjustments to better represent typical

Arctic processes (increased fresh snow density, reduced compaction within the canopy and lateral transport of snow by the

wind) have helped to improve the model outputs.

The significant amount of depth hoar and the Arctic-like density profile led to the conclusion that water vapor transport

was the dominant metamorphic process at TUNDRA. At FOREST, the density gradient is towards slightly lighter layers at

the top, indicating a role in overburden compaction. However, simulations indicate a significant role of water vapor transport,

showing that even thick moderately cold snowpacks in the low-Arctic significantly differ from alpine-type snowpacks. Thus,

after observing that water vapor transport is a crucial process in thicker Arctic snowpacks and the thick, moderately cold

snowpack in forested environments, we conclude that it is a critical process for the majority of seasonal snowpacks on Earth.

Thus, the integration of water vapor fluxes in snow models, particularly in those coupled to climate models is a pressing issue.

*Code availability.* The source files of SURFEX code are provided at the Git repository (http://git.umr-cnrm.fr/git/Surfex_Git2.git, last access: 15 November 2021) with several code management tools (history management, bug fixes, documentation, interface for technical support, etc.). Registration is required; a description of the procedure is described at https://opensource.cnrm-game-meteo.fr/projects/snowtools/wiki/Procedure_for_new_users (last access: 15 January 2022).

*Data availability.* Data sets allowing the driving and testing of snow models at the TUNDRA and FOREST sites are being prepared and will
be available soon.

*Author contributions.* GL, DN and FD designed research. GL, DN and FD deployed and maintained instruments. GL and FD performed the field work. GL analyzed data with inputs from DN and FD. GL set up the snow model Crocus with help from ML and MD. GL wrote the paper with inputs from DN and FD and comments from MD and ML.





*Competing interests.*  Florent Domine and Marie Dumont are Editors for The Cryosphere.

*Acknowledgements.*  This study was funded by Sentinel North, a Canada First Research Excellence Fund, under the theme 1 project titled,
"Complex systems: structure, function and interrelationships in the North". The authors are grateful to Denis Sarrazin for technical support
of the CEN weather stations and the community of Umiujaq for permission to conduct our research.This research has been supported by
the Natural Sciences and Engineering Research Council of Canada (Discovery Grant), the French Polar Institute (IPEV) (grant no. 1042)
and the Horizon 2020 program of the European Commission (INTAROS; grant no. 727890). CNRM/CEN is part of Labex OSUG@2020
(ANR-10-LABX-0056).



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
