# Peer review of "Snow properties at the forest-tundra ecotone: predominance of water vapor fluxes even in deep, moderately cold snowpacks"

_The Cryosphere, 2022_

## Referee Comment (RC3)

[referee-annotated manuscript omitted]

---

## Author Comment (AC1)

**Responses to Reviewer #1**

**General Comments:**
This paper addressed the snow properties such as snow height, stratigraphy, thermal conductivity, and density through observations at two sites, located in tundra and forest environment, in northern Canada. The observation results showed some contrasted properties between tundra and forest:

1. the higher snow height in the forest whereas lower height in the tundra.
2. lower density at the top of snowpack in the forest whereas homogeneous density profile vertically in the tundra.
3. the warmer temperature at the bottom of the snowpack in the forest than that in the tundra.

The authors furthermore proposed some modifications on parameterizations for snowfall density, viscus compression, and blowing snow implemented in the Crocus model. Then, the authors demonstrated the Crocus simulation with the proposed modifications reproduced better snowpack properties observed in the two sites than the default Crocus simulation. The content of this work is suitable for the scope of The Cryosphere.

Thank you for your comments on our manuscript.

However, at present, there are many problems for the paper to be published in 'The Cryosphere'. My general concerns are mostly as follows.

1. The title is far away from the subject defined at the end of the introduction. The title or the subject defined in the introduction should be revised. Moreover, the conclusion should be written as an answer responding to the subject.

The subject at the end of the introduction will be revised to align the research question with the title:

'Here, we present data on the internal physical properties of subarctic snowpacks to show that the transport of water vapor is an important process shaping the vertical snow density profile in both tundra and forest dominated areas. Furthermore, we test the performance of the snow model Crocus and explore adjustments to Crocus that compensate the lack of a water vapor transport mechanism in the model.'

2. The authors concluded that a vertical profile of snow density in the Arctic region is formed with a water vapor transport process rather than a viscus compression process. However, the physical logic to reach this conclusion is not enough given in the main text. In this study, the authors performed a numerical experiment using the Crocus model with parameterizations, developed focusing on snowfall, compaction, and blowing snow. Within this experiment, the water vapor transport is not updated.

Sublimation is also ignored. Nevertheless, the authors concluded that the density profile is dominantly formed through the water vapor transport rather than the viscus compression based on a result of numerical experiment and fact that depth hoar is predominantly observed at the bottom of snowpack in the study sites. However, because the numerical simulation is successful using parameterization modified by decreasing overburden at the bottom snowpack layer (due to vegetation), it is, rather, a more straightforward story that the density profile is a result of vegetation. Or, it is also potentially nice that the density profile is formed by increasing viscosity of snow layer where depth hoar is predominantly formed because the decrease in the overburden is practically the same as the increase in viscosity. Anyway, the authors should revise the logical process carefully to reach the conclusion based on the obtained results.

The conclusion that the vertical density profile in the Arctic is formed due to the transport of water vapor is not only based on the numerical experiment we performed. We discussed the observed density profile, thermal conductivity and the stratigraphy. All of those variables suggested that this process is important for shaping the Arctic snow cover (e.g. the abundance of depth hoar crystals at the bottom of the snowpack which arise only from water vapor transport and a density profile with relatively light snow at the bottom of the snowpack which is not expected by a compaction driven density evolution). In section 4.3, we discuss that observations lead to the conclusion that water vapor transport is the dominant process at TUNDRA and at least deemed important at FOREST. The numerical experiment was performed to back up the conclusion from the observations and show that the current processes implemented in the model are not sufficient to reproduce observations. Thus, we test a simple way to improve the model without taking the transport of water vapor explicitly into account as this is complex to implement. We clearly state that our improvements compensate for the lack of this transport (see L. 14 and 390)

Also, please note that sublimation is included in the simulations. We merely stated that we did not use an option to further increase it (L.108 in the manuscript).

3. Topic sentences are often absent or inappropriate. Also, multiple topics are sometimes included in a paragraph. The topic sentence is generally very important for readers to understand the paragraph smoothly. Moreover, because the multiple topics often confuse readers, the paragraph should be constructed with only one simple topic. The paragraphs that needed to be updated are pointed out in specific comments.

Please see our answers to comments on the specific paragraphs below.

4. The scientific originality is not enough or unclear at present. While the authors conducted many diligent in-situ observations for many years, their conclusions seem very similar to the previous works described in the introduction. For example, your result of 'higher/lower snow height in the forest/tundra' seems very similar to your introduction in L21–37. Moreover, the homogeneous vertical profile of density in the tundra is easily expected because a strong wind often makes wind slabs at the top of the snow cover. In the introduction section, the authors need to more clarify what the previous works addressed/found and what the lacking understandings are. Then, a single concise subject of this study should be defined at the end of the section. Moreover, in the discussion section, the new findings obtained in this study should be more emphasized. By the way, the Crocus simulation was interesting for me. As the authors pointed out in the main text, it is basically difficult to apply the current physical snowpack models to the Arctic region. Even with very simple modifications from previous works

(Barrere et al, 2017; Royer et al. 2021b), the model successfully simulated the observed snowpack properties, which should be more emphasized. However, I was a little bit disappointed that many concepts and evidence that were necessary to understand the modifications were omitted in the current manuscript.

The issue of water vapor transport and particularly the question of how to simulate this process is very timely and has been the subject of active research only recently (Fourteau et al. 2021, Jafari et al. 2020, Jafari et al. 2022, Simson et al. 2021). So we think that observations in a region where very few data exist, here the low Arctic tundra and open forest at the forest-tundra ecotone, are a valuable enrichment for the scientific community.
It is probably an exaggeration to mention that a homogeneous density profile is 'easily expected' on the tundra because of winds. Winds also affect snow at the beginning of the snow season. Why then do not they lead also to dense wind slabs? Clearly, there is no objective reason to "easily expect" our observations. In this specific case, it has been shown (Domine et al. 2016) that the densification of the top of the snow cover arises also from water vapor transported upwards and not just by wind packing. In the introduction we clearly mention studies that have simulated Arctic snow. However, there are no studies about detailed observation of the density and thermal conductivity in the forest-tundra ecotone. Concerning the research question, please see comment 1.

We are open to include the concepts and evidence required to better present the changes we have introduced, but this would require specifying more clearly what you are referring to.

**Specific Comments:**

5. L26–27: This seems important motivation for this study. The considerable changes described by Payette et al. (2001) and Ju and Masek (2016) should be introduced in detail.

Thank you, that is an excellent point. We propose the following addition:

"In their study, Ju and Masek (2016) found that 29.4% of the land cover in Alaska and Canada showed greening trends. According to their findings, the greening occurred primarily in the tundra, with Quebec and Labrador being the most affected regions. Considering these significant changes, the extent of the forest-tundra ecotone throughout circumpolar regions and its role in the global climate, more research is essential."

6. L42: What does 'this type of snow' indicate? Please clarify it.

We will substitute 'this type of snow' by 'Arctic snow'.

7. L44–46: This statement is inappropriate because the dominant process controlling the density profile is independent of the circumstances in the model development.

In the model development, the dominant process does matter as one might only include processes important for the setting where the model is deployed. Crocus was developed for avalanche research in the Alps. It only describes processes that are prevalent in alpine snow. In alpine settings, the compaction due to the overburden is far more important than water vapor fluxes. In Crocus these fluxes are omitted without detrimental impact on the performance of the model in this setting. The situation is clearly different in settings where water vapor fluxes is important.

8. L40–52: This paragraph probably contains two topics, so I suggest splitting this paragraph into two: the difficulty for modeling the arctic snowpack by the current default snowpack models and the current attempt to overcome the difficulty. The former can be reconstructed from the content of the current paragraph probably. For the latter, the authors need to describe the contributions by Barrere et al. (2017), Gouttevin et al. (2018), and Royer et al. (2021b) appropriately. Then, the unresolved problem should be given.

It is true that the paragraph could be divided into two paragraphs, but our initial idea was to dedicate one paragraph to observations and one to models. We believe that this is the best way to maintain a logical sequence that channels us towards the objectives of our paper.

9. L46: 'this deficiency' is unclear. Please clarify it.

We will replace 'this deficiency' by 'the lack of water vapor transport'.

10. L46–48: I understand modifications by increasing the maximum density are necessary for modeling the Arctic snowpack. However, this statement is a little bit confusing because the authors say the mismatch of density is due to the lack of consideration of water vapor fluxes in the model at the beginning of this paragraph. Please get the difficulty for the modeling arctic snowpack straight more.

This is a good point. In Arctic snow covers, water vapor is transported from the bottom to the top, effectively increasing the density at the top and decreasing it at the bottom (Domine et al. 2019). As this process is not present in the models, another way needs to be found to increase the density at the top, which was done by increasing the maximum density of wind-induced snow compaction. Thus, the lack of water vapor transport and the increase of the maximum density are directly related as suggested in the text. In other words, our strategy is to use an error compensation scheme rather than directly treat water vapor fluxes, as this would be much more complex as it would require a novel model architecture. This is a common strategy in model development/modification.

11. L52: This statement is confusing. Before this statement, the authors say the density in the forest is mainly controlled by the compaction as well as the alpine snowpack (L37–38). If so, because the models have been well validated on the alpine snow, readers intuitively expect the snowpack models can reproduce the density well even in the arctic forest. If the authors would like to problematize it

here, in my opinion, the authors need to describe reasonably how valuable the test to check the ability of models to adequately simulate density is.

This is an important subject of the paper, to find out whether the model is capable of simulating the snowpack in boreal forest environments. Studies on high Arctic tundra have shown that the model was unable to simulate the snow there (Barrere et al. 2017, Gouttevin et al. 2018). It is therefore not surprising that at TUNDRA, a low Arctic site, the model does not work either. However, at FOREST, the important accumulation makes this site more similar to an alpine site, so there is an interest to know how the snow model Crocus, developed for alpine applications, performs in this environment. As mentioned before we, will revise the end of the introduction where we state more clearly the research question and its importance.

12. L64–66: For this, a topographic map is suitable. Could you add such kind of figure?

A topographic map is included in the paper cited in L. 71, Lackner et al. (2021), in which a more detailed description is available. Furthermore, Figure 1b) gives a good overview of the valley and its topography. Moreover, the vegetation transition is clearly visible, in contrast to a topographic map.

13. L66: '70 to 80 % of the upper valley' is unclear because the spatial region of 'upper valley' is undefined. Within a topographic map (my comment #8), you can depict the region where the upper valley is. Or, 'the upper valley' should be replaced with 'the tundra'.

Indeed, it is difficult to define exactly the region of the upper part or the tundra as the regions and biomes merge into each other. We will add a line indicating the approximate limit of the upper part in Figure 1 in the manuscript.

14. L67: Similar to my comment #9, the region of the 'lower valley' is undefined.

See answer to comment 13.

15. L69: Is '2-3 m' a typo for '2–3 m'?

Yes, thanks, this will be fixed.

16. L86: In my opinion, the error of 29% is large, which potentially affects the conclusion. Despite that, this problem is not well discussed later. The authors should demonstrate that the conclusion would be robust even if the measurement included this magnitude of the error.

Yes, an error of 29% is large. We will include a discussion of the impact of this error at the end of section 4.1:

'Generally, the error of 29% for the thermal conductivity measurements is rather high, however, in this study, the gradient of the profile is our main focus, reducing the impact of the uncertainty on a single measurement. Moreover, by taking the average of all samples, the uncertainty is reduced according to $n^{-1}$ or a factor of $\approx 4$ with $n = 21$ (TUNDRA) or $n = 15$ (FOREST) samples.'

17. L86–87: What was the time-lapse camera used for?

The time lapse cameras were used to qualitatively observe the transport of snow and to check to which extent the snow poles were covered.

We will include this phrase in the manuscript.

18. L88–89: When did you conduct snow-pit observations eventually? Please add a table describing the dates of observations.

Good idea, we will add a table to the supplementary material containing all dates of the snowpits.

19. L91–91: Please describe the vertical intervals in the measurement of density and temperature.

The spacing varied between the snow pits. We mostly used a 3-cm spacing, but sometimes the spacing was increased to 5 cm. We will mention this in the manuscript:

"The spacing was mostly 3 cm between measurements but was increased to 5 cm for some snow pits, mostly the deepest ones."

20. L92: 'some snow pits' is unclear. Could you specify how many snow-pits is?

We will specify this in the table requested in comment 18.

21. L103–110: The topic of this paragraph is unclear. Probably the most important prior information for readers is that some parameterizations implemented in the Crocus are modified from the default settings. Moreover, this paragraph contains general reviews about the difficulty of modeling the arctic snowpack, which should be described in the introduction (see also comment #4). Please rewrite this paragraph and reconsider the appropriate topic sentence.

As shown by comment 22, the implementations for blowing snow might be a bit counter-intuitive, thus, we think that a quick reminder that the 1D nature of the model causes problems for simulating blowing snow is appropriate.

22. L104–106: Is this really true? The effect of blowing snow on snowpack is generally separated into erosion and accumulation (, and sublimation sometimes).

In reality, it is true that blowing snow is associated with erosion and accumulation, however, as we stated in the precedent sentence it is difficult to simulate those processes due to the 1D nature of the model. So alternatives were implemented to take into account some of the impacts of blowing snow. We will clarify this by explicitly stating the problem in the precedent sentence:

'Due to the 1D nature of the model, simulating blowing snow is problematic**, and snow erosion and accumulation cannot be simulated explicitly. Therefore**, two separate processes [...]'

23. L107–108: Could you add a quick description of the parameterization of Gordon et al. (2006) here?

The parametrization of Gordon et al. (2006) is not used in this study, so describing it here in more detail does not contribute to the understanding of the model used for the study.

24. (1): Do you mean that the viscus compression is ignored at the top layer?

No, this compaction is additional. We will clarify this:

[...] the upper snow layers are **additionally** compacted according to the following equation [...]

25. L102: Is 350 kg m-3 a value of the default Crocus?

Yes, it is. We will clarify this:
'(set as 350 kg m$^{-3}$ **by default**)

26. L115–122: This content should be described in the introduction. Moreover, I am wondering why the authors do not focus on the water vapor flux which is emphasized as a key process reproducing density profiles at the Arctic region in the introduction.

This is also described in the introduction (see L 41-43), here we repeat the motivation for implementing the modifications.

We did not focus on the water vapor flux as it is complicated to implement in a model. This is because water vapor transport requires the simultaneous transport and mass and energy which makes it difficult to implement in the current structures of snow models. Also, the main process transporting water vapor from the bottom of the snowpack to its upper layers is convection, but it is not possible to explicitly model convection with phase change in a 1D model.  So we tried to compensate the lack of this process by using other, simpler processes.

27. L123–124: This topic sentence is inappropriate for the content of this paragraph. The most important information is that the authors basically selected the parameterization of Vionnet et al. (2012)

as well as Royer et al. (2021b). Then, the description of modifications, specialized for your study site, to increase the density at the top of the snowpack should be given.

We will merge the paragraph beginning at L. 123 with the precedent one as there it is stated that our modifications are based on the parametrization in Vionnet et al. (2012) and Royer et al. (2021b).

28. L123–127: This is verbose. Please make the statements shorter.

We will split the sentence starting with "All three depend on..." in two, thank you.

29. (2): T_fus can be replaced with 273.15.

It could be replaced indeed, but we took the formulation from the original paper Vionnet et al. (2012) to avoid possible confusion when comparing both papers.

30. L130–132: According to the introduction, the hard slab at the top snowpack is induced by the strong wind, not the heavy density of fresh snowfall. Why is this modification appropriate for your study site?

The heavy density of the fresh snow is due to strong winds. Thus, the modifications take into account both, the densification of the snow while the crystals are still airborne (fragmentation of the crystals during saltation) and when they are already deposited on the snowpack. We will add a sentence to clearly state that our parametrization of fresh snow also takes into account the densification after deposition:

'Note that this parametrization includes densification effects of the wind during periods when the snow crystals are still airborne (fragmentation during saltation) as well as when they are deposited onto the snowpack.'

31. L132–133: Please describe this sensitivity analysis. Moreover, related to my comment #26, the density of the upper snow layer is a result of a fresh snowfall, sublimation, and wind-induced compaction. I am concerned that the selected parameters based on this sensitivity analysis are far away from the appropriate settings for the density of fresh snowfall in your study site.

Thank you. We will further describe the sensitivity analysis:

"These values were obtained with a sensitivity analysis where **we varied $c_\rho$ in order to obtain a good agreement between the simulated and observed densities of the top of the snow cover.**"

As mentioned in comment 30, the process takes into account both densification of fresh snow and when it is deposited on the snow cover. We do not give any recommendation to use the parametrization given here as a general equation to calculate fresh snow density. We just say that for our model this equation gives good results in terms of the initial snow density. Other users might want to change the process of snow densification of deposited snow and in this case clearly a different parametrization for fresh snow needs to be used.

32. L133–136: This statement is for the effect of wind-induced compaction and blowing snow, not for the density of fresh snowfall. If the authors say that the density of fresh snowfall is generally higher due to fragmented snow with stronger wind, it is an acceptable statement.

To actually consider the processes involved here a 2-D model would be needed where snow can be transported explicitly. However, since this is a 1-D model, it cannot actually describe the physical processes, it can only give a simplified numerical description whose aim is solely to reproduce observed densities. We thus simulate (i) the density of precipitating snow under windy conditions; (ii) the modifications of the density of surface snow due to erosion/deposition processes, under a 1-D simplification scheme.

33. L135: The vegetation height is undefined yet (, but approximate values are given in Section 2.1). Please specify its value. Moreover, is the vegetation height really appropriate, not the canopy height?

Here the exact value is not needed, we present merely the dependence of the parametrization on the vegetation height. However, we will change section 2.4 from just 'Forcing Data' to 'Forcing Data and Model Setup' where we will clearly state the heights used, 0.4 m for TUNDRA and 1.3 m for FOREST

34. L137: What is the stabilizing effect? Please describe it.

It is the effect that vegetation stabilizes the snow cover, effectively reducing compaction. We will add this:

'This takes into account the stabilizing effect **of** vegetation **on the snow cover** and [...]'

35. L139: Is D really snow density? It seems the thickness of the snow layer. By the way, snow density is already defined as $\rho$ (Eq. 2).

Thanks for the remark. *D* is indeed the layer thickness. We will correct this.

36. L142–143: A physical process represented by a factor c is unclear. Is c a fraction of overburden weight undertaken by the shrubs within the snowpack?

Yes, *c* is basically a factor that reduces the overburden. This will be stated in the manuscript:

"Herein, we introduce a parameter c acting **to reduce the effective overburden and as such, the compaction.**"

37. L143–145: Please describe this comparison. Is the stabilizing effect obviously observed in this comparison? How did you recognize the stabilizing effect from the observed density profile?

We will further explain the comparison:

"This value was obtained by comparing observed and simulated density profiles **and varying *c* until a good agreement with observations was obtained.**"

As detailed in comment 2 and 26, we were trying to compensate the lack of water vapor transport. Thus, we were seeking a process that produces the smaller densities at the bottom observed in the snow pits.

38. L147: The height of the canopy is undefined. Please specify its value. Related to my comment #29, is the vegetation height is more suitable?

We will specify the vegetation heights in section 2.4 which are 0.4 m for TUNDRA and 1.3 m for FOREST

39. L148: Please replace 'the lack of blowing snow scheme' with 'the lacking consideration of a blowing snow process' or the other appropriate one. According to the authors' expression earlier, the blowing process is implemented as sublimation in the Crocus.

Thank you, we will change the sentence as suggested.

40. L152–153: What are offline simulations? Does not the Crocus interact with a parent land surface model?

As described in L. 96, Crocus is part of SURFEX. Offline simulations are simulations where the land surface model is used without an atmospheric model e.g. the forcing has to be provided externally as we have done it here.

We will replace "For offline simulations" by "For offline simulations (no coupling to an atmospheric model)".

41. L155: According to L160–161 and Eq. (4), the minimum value of (a+bW_s) is 1.0 at W_s=3 (m/s), meaning P_new=P_old. So, how did you remove snow without changing sublimation?

With this statement, we refer to the fact that when changing the precipitation with equation (4), one can remove or add snow. In our case, we do not remove any snow, but one could do so by changing the parameters *a* and *b*, for instance at sites where snow erosion is clearly visible.

42. L156: Please remove PR because this abbreviation is not used anywhere else.

Thank you for the comment, PR will be removed.

43. (4): Is the formulation of a one-order linear equation really appropriate in order to account for the blowing snow effect? Could you add appropriate references?

It is true that blowing snow effects are very complex. In L. 382-386, we advocate that it should get increased attention in the Arctic. Simulating blowing snow in a 1D model at the site scale only is inherently very difficult as the eroded snow at one point needs to be accumulated at another point. In L. 151, we say that not including any process of this kind greatly impacts the simulation (see Figure 9 and 10). Thus, in order to test the other processes, we needed to take blowing snow into account.

44. L158: Please add a unit for P_new and P_old.

Units of P_new and P_old will be added, thanks.

45. L158: Is P_old the observed precipitation rate, corrected using a transfer function (L183), at TUNDRA?

Yes, P_old is the initial rate as given to the model through the forcing data which contains the corrected precipitation at TUNDRA.

46. L159–160: This pre-analysis should be given as supplemental information, at least.

As stated in L. 167 the values are not optimized and in L. 382-389 we state that we did not aim to find the best set of coefficients but rather emphasize the importance of some processes. Including an analysis would give the impression that we recommend the used parameters for other sites and/or models, which we do not wish to do. For this reason, we feel it is best not to include this specific analysis.

47. L161–162: If so, there is a large gap in the increase in precipitation around 10 m/s wind speed because (a+bW_s)=3.1 at W_s=10 in Eq. (4). Does not this gap affect the result of the Crocus simulation?

Thanks for pointing out the error. We did not limit it to twice the original precipitation rate but to the 3.1 mentioned here. This error will be corrected.

48. L163–165: This preliminary series of tests should be given as supplemental information. By the way, are the preliminary tests really necessary? In this study, the Crocus simulation is performed on only two sites, and precipitation is observed at the TUNDRA site. Therefore, the necessary preprocessing is to estimate precipitation at the FOREST site including the blowing snow effect.

Preliminary only refers to the first test with Crocus and their analysis, so basically the default version described in the paper, thus, these tests are already included. There we saw that at TUNDRA no blowing snow was necessary but only at FOREST.

49. L165–166: Is this really true? As I pointed out in comment #37, Eq. (4) has only an effect to increase precipitation.

Yes, you only need to make the parameter b negative and than equation (4) can be used to subtract snow.

50. L171–172: What is the time interval of forcing data? And please remove '(solid and liquid)'.

The time step was one hour. This will be clarified:

'**Hourly** observations of these variables at each of the two sites have been collected since 2012,[...]'

It is important here to mention that we forced the model with solid and liquid precipitation as there are also models which can only take precipitation and then partition it internally using some sort of threshold.

51. L174: Which grid point of ERA5 did you select?

We selected the point closest to the location of the site which is 76.5º W and 56.5º N.

52. L177–181: These statements are not for the forcing data, but for the model settings. Please move to an appropriate position. Related to this, please place an appropriate topic sentence at the top of this paragraph. Moreover, what are the initial temperature and the bottom boundary condition?

Indeed this statement is not concerning the forcing data. We will change the section's name to 'Forcing Data **and Model Setup**'.

The initial temperature does not matter as we used a 5-year spin-up to ensure thermal equilibrium. At the bottom the heat flux is set to 0, which will be stated in the manuscript.

53. L187: However, the authors do not describe the difference of forcing data between TUNDRA and FOREST in this subsection. Please revise this topic sentence appropriately.

Sure, thank you. We will revise the paragraph as follows:

"We used a different forcing data set for each station. However, some of the required variables were not available at FOREST, so we used the precipitation, pressure and specific humidity from TUNDRA. Given the proximity of the two sites, the differences between these variables are presumed to be very small."

54. L192: Please concatenate this paragraph with the next one.

We will concatenate the two paragraphs.

55. L203: interannual variability?

We will change "fairly variable interannual amplitudes" to "pronounced interannual variability".

56. Figure 3: A color of 2019-20 is different from that of Figure 4.

We will change the color of 2019-20 in Figure 3 to correspond with the one in Figure 4, thanks.

57. L208–209: Please concatenate this paragraph with the next one.

We will concatenate the two paragraphs.

58. L210: However, a depicted line for 2015/16 at the FOREST site begins at the end of Nov. How did you recognize the onset of snow cover?

Unfortunately, the instrument monitoring snow height at FOREST had several malfunctions. This plot only depicts good quality data, so for the year 2015/16, it was not possible to detect the onset of the snow cover with snow height observations. We were, however, able to estimate it thanks to time series of upwelling shortwave radiation. This detail will be included in the figure caption.

59. L222–223: How many samples did you take?

At this instance we took 172 samples. We will indicate the number in the text:

'For instance, on 12 April 2018, we made 172 measurements of the snow height within a 100 m radius of TUNDRA [...]'

**60.** Figure 5: This figure is not suitable for a scientific paper because the figure only shows a result through unclear/subjective post-processing by authors. At least, the source results of the observed stratigraphies should be given as supplemental information.

We will include the observed stratigraphies in the supplementary material.

**61.** L239: How was the mean calculated? According to Fig. 6, the vertical positions of each measurement were different, so the mean value of the vertical profile would not be simply obtained.

As seen in Figure 6, the height was normalized and subsequently we calculated the mean density in each bin of normalized height.

**62.** L241: How did you normalize the vertical scale? Simply did you divide it by the height of snow-cover? I suggest normalizing the vertical scale with a logarithm.

Yes, we divided every measurement height by the height of the respective snow cover. We do not see the advantage of a logarithmic scale in this context. Using the snow heights discussed in section 3.2.1, one can very simply get an impression of the intermediate snow heights with a linear scale as opposed to using a logarithmic scale.

**63.** L242–248: However, there are very large variabilities among the profiles. I suggest depicting confidence intervals in the figure and verifying robustness.

By showing a spaghetti plot as we do here, we believe we can illustrate the range of variability from one profile to another. Overlaying confidence intervals would, in our opinion, make the graph difficult to interpret. In order to assess the robustness of presenting only the mean, we tested the idea of representing the median of the distribution. As shown in the figure below, we can see that the two curves are very similar.

[Figure]

[Figure]

Figure 1. Snow density profiles from 29 snow pits in tundra environments and 18 snow pits in forested environments collected between January and March from the years 2012 to 2019. For better comparability, snow heights were normalized. The means and medians of all profiles are also shown.

64. Figure 7 and L249–257: Same things as comments #57–59.

Please refer to the figure below. Our statement from the previous comment stands.

[Figure]

[Figure]

Figure 2. Same as Figure 6 but for snow thermal conductivity. Note that only data from 21 snow pits for TUNDRA and 17 for FOREST were used, as thermal conductivity measurements were not collected for every snow pit.

65. Figure 8: In the caption, there is a statement of 'Heights at which measurements were taken are relative to the surface of the ground'. On the other hand, in the legend, the unit of height is cm, not relative value to the surface of the ground. So, what are the heights depicted in the figure?

As mentioned in the caption, the heights are given relative the ground surface. This means that the ground surface is at 0 cm. Consequently, –9 cm means 9 cm below the surface and 11 cm means that it is 11 cm above the ground surface. Note that in the revised version we will change the height indications at TUNDRA, as we change the point of reference from the soil-lichen interface to the surface of the lichen. This will be clearly stated in the manuscript.

66. L268–269: From mid-Mar., the height of snow-cover is exceeding far away from 53cm (Fig. 4). How did you recognize the temperature of the top snow layer?

We will rephrase the sentence and no longer use top layer:

"At TUNDRA, the difference between air and the temperature at 64 cm varied between 0º and 5ºC".

67. L269–270: Same as comment #62. From mid-Jan., the height of snow-cover is exceeding far away from 64cm.

We will change the sentence to:

"At FOREST, the difference between air and the temperature at 64 cm reached up to 10ºC, illustrating the impact of the greater snow depth on the snow temperature."

68. L292–295: This result and L159–160, where the authors say a reasonable agreement between the simulated and observed snow height, contradict each other. Is the obtained parameters a and b in Eq. (4) really appropriate?

The statement in L159–160 refers to the general snow height, which is reasonably simulated, as can be seen in Figure 9.  L292–295 are about specific events where inconsistencies arise due to a lack of a proper blowing snow scheme. This, however, does not strongly affect the overall snow height and equation (4) is not used. Also, here we talk about TUNDRA and earlier we state that equation (4) is only used at FOREST.

69. L295–296: Probably this is true. However, in order to say this, ideally, the snow water equivalent should be checked because the snow height is a result of snowfall, compaction, blowing snow, and sublimation. Could you add such kind of figure?

Thank you for this excellent suggestion. We will add a figure depicting the SWE in the supplementary material. See answer to comment 2 of reviewer 2.

70. L298: Please concatenate this paragraph with the next one.

We will concatenate the two paragraphs.

71. L305: The 'residual' is unclear here. What is the residual from?

We will change 'residual' to 'difference'.

72. L311: Same as comment #67. The residual is unclear.

We will change 'residual' to 'difference'.

73. L326–334: Does sublimation, ignored in this study, affect the contrasted snow height between TUNDRA and FOREST?

We do not ignore sublimation in this study.

To clarify this, we will add the following phrase:

'In other words, sublimation is included in the study, but it is not increased by the blowing snow parameterization.'

We just chose not to further increase it, as measurements from Lackner et al (2022) suggest that simulated and observed sublimation are already comparable. Although sublimation is likely to differ between the two sites, judging by the generally low rates, we do not believe it plays a significant role in the snow height discrepancies.

74. L332–333: I miss this observation result in the result section. Is this the snow height at TUNDRA?

No, this is not at TUNDRA. However, 'very top of the valley' might be confusion here. We will clearly state that the observations were taken further up the valley from TUNDRA:

"Snow surveys that were conducted at the very top of the valley **(≈ 500 m from TUNDRA)** revealed a very thin snowpack (<40 cm)."

75. L333–334: This sentence seems not to be related to the blowing snow effect. Is this really necessary to emphasize the effect of blowing snow?

The fact that the snow height at TUNDRA is more closely correlated to precipitation than at FOREST is an indication that blowing snow does play a significant role for the snow height at FOREST, whereas its role is less pronounced at TUNDRA.

76. L348: This is an inappropriate topic sentence. Probably a key point of this paragraph is that the mismatch between density and thermal conductivity profiles is not simply explained by the traditional relationship. Please revise it.

We will revise the topic sentence:

"A key factor that is not included relationships between snow density and thermal conductivity (e.g. Sturm et al. (1997), Calonne et al. (2011), and Fourteau et al. (2021)) is the snow type."

77. L350–351: Evidence is obviously lacking for this hypothesis. At first, you need to demonstrate how much the traditional relationship between the snow density and thermal conductivity explains the result

of this study. Then, a potential reason for the mismatch should be given. Appropriate references, that show a correlation between the thermal conductivity and snow grain shape, are also necessary. Otherwise, this paragraph should be deleted.

Thank you for raising this important point.  For instance, Lehning et al. (2002) discuss the impact of microstructure (i.e. snow type) on thermal conductivity. We will add this reference in the text.

78. L357–359: This is an inappropriate topic sentence. Probably the content of L362–363 is a suitable topic for this paragraph.

This sentence states that simulating Arctic snow is generally a challenge as most sophisticated snow models have been developed for alpine snow. We believe it properly introduces the content of the paragraph, and as such wish to keep it as is.

79. L357–371: This paragraph is redundant. Please make the paragraph shorter. Moreover, please demonstrate how your modifications from Barrere et al. (2017), Gouttevin et al. (2018), and Royer et al. (2021b) improved the simulation skill.

The setting is different, we use these modifications in the forest-tundra ecotone, whereas the other studies were done in the Arctic. Then, we detail that we change the impact of vegetation.

80. L375: non-linear equations?

The study proposes allometric equations.

81. L372–378: So, did your implementation, not taking the whole vegetation height as a zone where compaction is reduced, effectively improve the simulation score? Please demonstrate it quantitively.

As stated earlier, we did not focus on a quantitative assessment, but rather by exploring possible means to improve snow simulations in the Arctic.

Furthermore, the choice to take a reduced vegetation height was based on the fact that shrubs bend (Ménard et al. (2014)). As such, we believe that taking the full height is physically less sound.

82. L390–392: The parameterizations implemented in this study are developed focusing on fresh snowfall, compaction, and blowing snow, not focusing on upward water vapor fluxes. Therefore, it is very hard to understand this sentence. Please update your statements.
Section 4.3: This section can concatenate with section 4.2.

Exactly, we argue that it would be necessary to include the water vapor transport in order to avoid site-dependent parameters. So it is the absence of this process in the model together with the fact that the

parameter are site-dependant that leads us to the conclusion that water vapor flux is important for Arctic and boreal snow.

As requested by the other reviewers, we will extend section 4.3. Thus, we won't merge this section with section 4.2.

83. L397–398: This is hard to understand. No modification accounting for water vapor flux was implemented in this study. Sublimation is also neglected. Nevertheless, why can the authors conclude that the water vapor transport is dominant over compaction? Moreover, it is unclear what kind of physical quantity is dominantly controlled by water vapor transport rather than compaction.

As also explained in the response to comment 2, we do not base our conclusion solely on the numerical simulations. In fact, a large part of this conclusion is based on the observations e.g. the stratigraphy containing depth hoar, the density profile and the strong temperature gradients which favour water vapor transport. For this conclusion, the numerical simulations serve as additional evidence. The high temperature gradients create water vapor transport and as this process is absent in the model this backs up the conclusion that water vapor transport is dominant in shaping the density profile.

To make it clear what kind of physical quantity is controlled, we will clearly state that density profiles are most affected and thereby also the thermal conductivity.

As mentioned before, sublimation is simulated by Crocus and included in the simulations.

**References**

[revised manuscript text omitted]

---

## Author Comment (AC2)

**Responses to Reviewer #2**

**General Comments:**

In "Snow properties at the forest-tundra ecotone: predominance of water vapor fluxes even in thick moderately cold snowpacks", the authors present an observation dataset and modelling with CROCUS of snowcovers in a forest-trundra eco zone in north eastern Canada.

Overall the manuscript is well written. The ability of modern snow models to accurately resolve snow depth and microstructure including accurate density estimates remains a key challenge. This paper presents some interesting observational and modelling results that, after tightening should make it a contribution.

Thank you for this positive appreciation.

1. My major criticism is that the treatment of the canopy and canopy impacts, e.g., canopy interception and canopy sublimation, are not clearly articulated. This is especially the case in the modelling description. I would like to see this more readily described. Without, it makes the forest results difficult to interpret. I understood the forest site to be similar to the Ménard, et al site where the shrubs are buried by snow (~1m tall), however the black spruce (~5m) are almost certainly not buried and thus have canopy affects. Given the canopy dynamics can impact shortwave transmittance, longwave (in/out), albedo, etc I feel the manuscript is missing this critical section.

As detailed in the manuscript (L. 68), trees only cover approximately 20% of the surface. Thus, there is no closed canopy, but rather sparsely distributed trees. Snow observations were made at some distance from the trees. As such, any impact of interception would be negligible. The impact of trees manifests itself on the reduction of surface turbulence, and hence of increased deposition of drifting snow and reduced snow erosion. We will clarify this by adding the following phrase after L. 94:

'At FOREST, snow pits were dug at some distance from trees, so interception of snow is negligible.'

The impact on shortwave transmittance is likewise neglected, given the precise location of our observations (no obstructions by surrounding trees) and the limited tree density. Observations were made where shrubs were present. Shrubs were treated as in the Tundra case, i.e. bending, densification. This will be clarified in the manuscript:

'At the exact location of the instruments at FOREST no trees were present that obstructed measurements.'

2. More of a "this surprised me" rather than a criticism, I was very surprised at the total lack of SWE observation and comparison with the model. I realize the focus of this manuscript is on the depth and density estimates. However, as snowdepth includes the uncertainty in both snow mass and density, it is

a bit difficult to attribute differences in SD as entirely due to density errors versus more systematic snow mass errors. For example, biases in snow loss due to surface sublimation cannot be diagnosed with snowdepth results alone. I would strongly suggest a small comparison of model v. obs SWE so-as to allow the reader to confirm snow mass is being correctly estimated.

Indeed, we chose to focus our analysis on snow depth and density, as well as on the vertical structure of the snowpack and its evolution in time. Obviously, as both the density profile and the snow height are computed, conclusions on the SWE can be drawn as well. As the number of measurements from which we can calculate the SWE is small and limited to one short period in each winter, it is impossible to follow the evolution of the SWE, for instance before and after major snow blowing events. For this reason, we decided to focus on the snow height to show the evolution of snow accumulation. However, we acknowledge the importance of SWE for hydrological applications. Therefore, we will add a figure depicting some SWE values calculated from the density profiles to show that in fact the SWE simulations are rather comparable to observations. We will include the figure in the supplementary material and we will mention it at the end of section 3.3.1 together with a short discussion of the results:

[Figure]

Figure 1: Observations and simulations of the SWE at TUNDRA and FOREST during winter 2018-19.

"Discrepancies in the simulation of snow height can also arise due to uncertainties of the SWE e.g. the total snow mass. To verify whether this is the case, we compared simulations of the SWE with observations that were obtained using the density profiles (see Supplementary Figure 1). At TUNDRA, there is a good agreement between the observed and simulated total snow mass. At FOREST, the model underestimates the SWE, similar to snow depth in Figure 9. Thus, the amount of blowing snow was higher than the one simulated by the model."

3. Lastly I think there needs to be a better treatment of the uncertainty of the parameters in the snowmodel and in the observations (specifically the conductivity). I realize the authors are not interested in calibrating the model. However, for example, how impactful are the decisions on line 160? Although I get the sense these are chosen by evaluating the physical processes in play, they are still somewhat arbitrary and may dramatically impact the interpretation of the results.

This is relevant remark. We will discuss the impact of the uncertainty of the thermal conductivity at the end of section 4.1:

"Generally, the error of 29% for the thermal conductivity measurements is rather high. However, in this study, the vertical gradient is our main focus, reducing the impact of the uncertainty of a single measurement. Moreover, by taking the average of the all samples, the uncertainty is reduced according to $n^{-1}$ or a factor of $\approx 4$ with $n = 21$ (TUNDRA) or $n = 15$ (FOREST) samples.'

and the uncertainty of the parameters at the end of section 4.2:

'The parameters in the processes *Compaction* and *Snowfall* were robust against small variations, meaning that the output of the model did not heavily depend on the exact choice of the parameters.'

However, we don't think it is useful to discuss the impact of the uncertainty of the parameters used in the process *Blowing Snow.* We clearly state in section 2.3 that this process was only implemented to increase the snow height at FOREST as otherwise simulations there would make no sense. We do not recommend the general implementation of the parametrization used in the process *Blowing Snow* in snow models and discussing this in detail could create the contrary impression.

**Specific Comments:**

5. L5 The TUNDRA and FOREST sites being all-caps surprised me. I'm fine with this, but I am wondering what the motivation is versus proper names such as "Tundra" and "Forest"?

We decided to capitalize TUNDRA and FOREST to make a clear distinction between the case when we refer to those sites in contrast to the tundra and forest biomes.

6. L16 "models leads to an inadequate representation" of snowdepth or SWE? The distinction matters w.r.t policy. E.g., if the mass is still right, then at these scales that is often sufficient: "we will still have X m^3 water input to reservoirs under policy Y".

Here, we refer to the density profile. This will be clarified:

[...] leads to an inadequate representation **of the density profiles** of even deep and moderately cold snowpacks, [...]

7. L34 "precipitation are typically higher" Given this study focuses on the transition, certainly they remain somewhat similar 1km apart. The boreal forest is a large region. Where, exactly, is this transition point?

We agree that in this case the precipitation is very similar. L.34 is a general statement about the distribution of precipitation in the Arctic versus the boreal forest. Groisman and Easterling (1994) showed that typically boreal forest environments receive higher annual precipitation than the Arctic tundra.

8. L38 "similar to alpine snow" This is below-tree-line alpine snow?

No, not necessarily, alpine environments usually feature an increased amount of snow and are not as cold as the Arctic tundra. This is why alpine snow covers show some resemblances with those in the boreal forest.

9. L 55 I would like to see the authors directly specify the research questions. This has a good start, but I would like this clearly stated and then answered in the discussion+conclusion

We will rephrase the last section of the introduction to make the research questions (formulated as objectives) more apparent:

"Here, we present data on the internal physical properties of subarctic snowpacks to show that the transport of water vapor is an important process shaping the vertical snow density profile in both tundra and forest-dominated areas. Furthermore, we test the performance of the snow model Crocus and explore adjustments to Crocus that compensate the lack of a water vapor transport mechanism in the model."

10. L 72 Figure 1 I really like this figure

Thank you. We appreciate your comment.

11. L 95 This section needs a description of how canopy interactions (mass + energetics) are handled

With the vegetation used here, shrubs, there is no canopy interaction in Crocus. The only dependencies of the snow cover on the vegetation are the ones introduced in this study and which are described as *Compaction, Snowfall* and *Blowing Snow*. We will state in the manuscript that no canopy interception is present in Crocus for shrubs.

12. L180 "based on estimates" are these from just musing on it, or were these informed from soil pits, etc?

These are estimates based on soil pits dug on several occasions. We will clarify this in the manuscript:

'[...] clay for FOREST based on estimates **from several soil pits dug around the station where higher fractions of fine particles where found compared to TUNDRA.**'

13. L181 Based on the met data availability I had expected a simulation period of 2012+ with a spin up period pre-2012. Why was the model not spun up prior to 2012 and run for evaluation 2012 onward? It would be good in this section to explicitly note "met data available for 2012-XXXX, model spin up was YYYY-ZZZZ, and evaluation was PPPP-QQQQ".

As stated in the manuscript, the precipitation data was available from 2016 onward (there is a typo in the manuscript, it has to be 2016 instead of 2015). Precipitation has a huge influence on the simulation, for instance when it comes to the compaction due to the overburden. For this reason, we did not use the

meteorological data prior to 2016 for the evaluation itself but for the spin-up. It is already stated that this spin-up was from 2012 to 2016 and we will add a phrase stating the start and end of the evaluation.

"[...] raw ERA5 data had to be used for the 2012–2016 period**, for this reason the evaluation period is from 2017 to 2020.**"

14. L182 Does this not contradict the 2012 statement on line 172? "Observations of these variables at each of the two sites have been collected since 2012, except for atmospheric pressure,"

Yes, indeed we will add precipitation as an exception as well.

"[...] except for atmospheric pressure, which was available since June 2017**, and precipitation, which was available since May 2016.**"

15. L182 "corrected the precipitation" is this the ERA5 data? Or the obs? Please explicitly state.

Only the raw precipitation data needs to be corrected for undercatch particularly of snow. We will clarify this:

"We corrected the **observed** precipitation data [...] "

16. L184 "fixed threshold of 0.5" There are a plethora of threshold methods that are physically based and indeed the choice matters significantly
    1.Harder, P. & Pomeroy, J. W. Hydrological model uncertainty due to precipitationâ      phase partitioning methods. _Hydrol Process_ **28**, 4311–4327 (2014).

    2.Jennings, K. S., Winchell, T. S., Livneh, B. & Molotch, N. P. Spatial variation of the rain–snow temperature threshold across the Northern Hemisphere. _Nat Commun_ **9**, 1148 (2018).

We agree that the choice does matter significantly. However, as we do not consider the snowmelt period, the threshold only affects the onset of the seasonal snowpack in fall, e.g., its lower layers. As the air temperature remains well below any reasonable rain-snow threshold during winter, the vast majority of the snowpack does not change when varying the threshold.

Furthermore, in L.185, we state that we obtained the threshold used in this study by comparing air temperature with observations on the type of precipitation from an observer at the nearby airport. The type of precipitation highly depends on quantities such as the atmospheric lapse-rate rather than on quantities measurable at surface level. Thus, we think that a threshold determined from several years of observations on the type of precipitation at the site is sufficiently robust.

17. This would be a good candidate for inclusion in additional uncertainty estimates to understand how impactful this was in the fall and spring seasons.

As mentioned in comment 16., spring is not considered in this study.  Concerning the impact in fall, we tested limits between 0.3 and 0.8 °C and did not see any significant impact on the snow quantity in fall. In the manuscript we will add this point:

'Thresholds between 0.3 and 0.8 °C were tested with little impact on the amount snow.'

**18.** An adjacent question is how was precipitation temperature estimated due to its impact on developing snowpack cold content?

In the model, precipitation temperature was taken to be equal to snow temperature. This will be stated in the manuscript:

'The temperature of new snow is taken to be equal to the temperature of the uppermost snow layer.'

**19.** L188 "specific humidity" this one may not be identical

 1.Flerchinger, G. N., Reba, M. L., Link, T. E. & Marks, D. Modeling temperature and humidity profiles within forest canopies. _Agr Forest Meteorol_ **213**, 251–262 (2015).

It is true that there might be differences in specific humidity, however, relative humidity was measured at both sites but unfortunately at FOREST there is only a short period when the instrument worked properly. Thus, we decided not to take those measurements for the model forcing at the FOREST site, but we used available measurements at FOREST to compare measurements at both sites (see Figure 2 below) and concluded they were sufficiently similar to use the same values for the forcing at both sites.

[Figure]

Figure 2: Scatter plot of the relative humidity at TUNDRA versus the relative humidity at FOREST for the period October 2015 to July 2016 when measurements at both sites were available. Also included are a linear fit with the corresponding equation.  This fit is very close to the 1:1 line, also plotted.

**20.** L198 "mean difference" And I assume gusts too, important for blowing snow

That is correct, for blowing snow gusts are important and just as the mean difference of the wind speed gusts are also lower at FOREST.

21. L199 "downwelling shortwave" is this sub canopy? Shrubs or in the forest proper? If the latter I would have expected substantially more difference late season. It may be worthwhile breaking this into a few periods as the long, dark winters will heavily bias the mean.

Both measurements were performed above the canopy (shrubs) and as the trees at the FOREST site are sparsely distributed, their impact on the radiation is likely not significant.

22. L201 "remained comparable throughout the winter" This seems expected due to low solar angles?

Indeed, this was somewhat expected.

23. L202 Or due to the canopy + higher solar angles?

As mentioned in comment 21., the canopy at FOREST is very sparse and does not have a strong impact on the downwelling radiation.

24. L205 Figure 3, note the site and add units.

We will add the site (TUNDRA) in the caption. However, units are already present (mm).

25. L217 "depending on the maximum snow height" isn't this somewhat a tautology such that deeper, colder snow packs take longer to melt out than small snowpacks? Noting the impact of cold content development might help make this section stronger.

Yes, it is expected that deeper snow covers take longer to melt. However, here we specify quantitatively what impact the increased snow depth has e.g. that the meltout date might be half a month earlier or later.

26. L241 "similar environments" unclear what this is referring to.

This refers to environments with a similar canopy.

We will explicitly state this and exchange 'similar environments' with 'environments with a similar canopy cover'.

27. L241 Figure 6 these captions are Proper Name case Tundra and Forest. Either change the text to proper name case or change these to TUNDRA/FOREST for consistency.

Thank you for the remark, we will change the captions to TUNDRA/FOREST.

28. L253 "for the upper 80%" It looks closer to 50-40%?

It is true that the decrease is less pronounced from 0.2 to 0.5 of the normalized height. We therefore change the phrase to:

'there was a clear decrease in density for the upper **50% and a very slight decrease between 15 and 50 %**'

29. 48 Figure 7 Suggest adding uncertainty regions for these observations to match the 29% noted in the text.

Our focus here is more on the gradient of the thermal conductivity which differs between TUNDRA and FOREST. As you suggested in comment 3., we will revise our discussion of the errors. There we will underline also that our focus is on the gradient rather than the specific value.

30. L281 Best remind the reader quick what was adjusted

We will state the processes included:

'Simulations with the adjusted version of Crocus **(including the processes Compaction and Snowfall)** [...]'

31. L283 "between the two versions" adjusted v. Non-adjusted?

Yes, this is what we meant. We will clarify this:

'One striking difference between the **adjusted and the default** version is [...]'

32. L286 Canopy impacts?

As we stated in comment 11., there is no canopy impact in Crocus prior to our adjustments with this type of canopy.

33. L295 Without doing a falsification experiment of one with and one without can you know this 100%. Please describe how it is known with such high confidence that it is related to these events.

In the model, snow accumulation can only be observed due to precipitation events. So, every rise in snow height is caused by snowfall. In reality, blowing snow can increase snow height even without a precipitation event or accumulated snow during a precipitation event can be eroded leading to a decrease in snow height. This was observed with time-lapse cameras which were installed at TUNDRA. So if there is an increase in snow height in the simulation (due to snowfall) but not in observations, the only possible cause can be erosion by blowing snow. Vice versa, if there is no increase in snow height in the simulations (and no precipitation)  but an increase in the observations, blowing snow is also the only possible reason.

34. L321 "as for the low[...]there." Is this not canopy as well?

No, as mentioned in comments 21. and 23., the forest is very sparse thus, the lower elevation relative to the ridge confining the valley has a greater influence than the trees.

35. L397 This water vapour transport finding seems to be a major conclusion I think you should better highlight

Indeed, we agree and will change the discussion to highlight this finding more:

'Hence, water vapor transport is the dominant process in shaping the density and thermal conductivity profile at our site, exceeding the importance of compaction due to overburden weight. Domine et al. 2016 and Domine et al. 2019 have shown this for thin high-Arctic snowpacks but here, snow depth is often more than 1 m and thus, considerably deeper than in the high-Arctic. Thus, water vapor transport is prevalent for all Arctic regions, even for those with relatively large amounts of precipitation.'

36. L403. Same as above

We agree that this is a major conclusion but the importance of it is already highlighted by the subsequent paragraph where we detail the impact of the finding for Arctic and boreal biomes.

37. L423 "Arctic-like" this is unclear, previously you had noted Alpine-like. Is that what you mean here?

Here, we refer to the snow at TUNDRA which is rather typical of the Arctic. The snow at the FOREST site is more alpine-like. We will clarify this in the subsequent phrase:

'At FOREST, the density gradient is towards slightly lighter layers at the top, indicating a role of overburden compaction more similar to alpine-like snow.'

38. L435 by publication-time I hope?

Yes, the data files will be deposited in a repository with a DOI before publication of this paper.

**References**

Domine, F., Barrere, M., and Sarrazin, D. Seasonal evolution of the effective thermal conductivity of the snow and the soil in high arctic herb tundra at Bylot island, Canada. *The Cryosphere,* https://doi.org/10.5194/tc-10-2573-2016, 2016.

Domine, F., Picard, G., Morin, S., Barrere, M., Madore, J.-B., & Langlois, A.  Major issues in simulating some Arctic snowpack properties using current detailed snow physics models: Consequences for the thermal regime and water budget of permafrost. *Journal of Advances in Modeling Earth Systems*, 11, 34– 44. https://doi.org/10.1029/2018MS001445, 2019.

Groisman, P. Y. and Easterling, D. R.: Variability and Trends of Total Precipitation and Snowfall over the United States and Canada, *Journal of Climate*, 7, 184 – 205, https://doi.org/10.1175/1520-0442(1994)007<0184:VATOTP>2.0.CO;2, 1994.

---

## Author Comment (AC3)

**Responses to Dr. Charles Fierz**

**General Comments:**

This contribution addresses the problem of water vapor fluxes modelling in snow-cover models, in particular in regard to Arctic snow and pertinently state on line 52, '... the ability of those models to adequately simulate density profiles has yet to be tested.'. For this purpose the authors use a consolidated multi-year data set collected at two nearby sites within the forest-tundra ecotone and model snow-cover evolution with the detailed physical snow-cover model CROCUS. To do so, three key processes for Arctic snow are adapted to get a much better representation of snow depth evolution as well as measured density and effective thermal conductivity profiles for these two sites.

Thus, even though some results maybe 'site-specific biased', the authors convincingly show that, '..., the integration of water vapor fluxes in snow models, particularly in those coupled to climate models, is a pressing issue.' (see line 429) while blowing snow is a further aspect that needs more attention but difficult to integrate in point simulations.

The paper thus addresses timely a need for improvement in snow-cover modelling and provides a comprehensive data set that can be used in future evaluations. Unfortunately, even so the text is well structured and generally pleasant to read, there are a number of issues I address in detail below.

Thank you for your encouraging words. We highly appreciate your comments, which will clearly help us improve our manuscript. Please find our answers to your comments below.

In summary I recommend accepting the paper after the authors addressed the issues below and the editorial suggestions found in the annotated manuscript.

Thanks for the detailed corrections in the annotated manuscript. We will implement all of them.

**Remarks on terminology:**

1. 'thick' vs 'thin': I think you need to make this distinction early in the paper and not as late as on Line 404. I would suggest to use deep and shallow. See title too, I stumbled over it.

We will follow your suggestion to use deep and shallow. However, in the introduction we already make the distinction between shallow Arctic snowpacks and deeper snowpacks in the boreal forest.

2. 'snow height' vs 'snow depth', Both terms are used in text and figures. Please use one only throughout the manuscript and I would prefer 'snow depth'. In figures do not capitalize the second word.

Thank you for the remark, we will use only snow height in the manuscript. The reason for that is, that when you make a snowpit, you consider snow from the base and record the height of a layer of interest, not the depth.

3. Line 13: '*overburden weight*' I wonder whether 'overburden' would not suffice, or at least switch to 'overburden load'?

We will change all occurrences of 'overburden weight' to 'overburden'.

4. Line 203: '*years*' I would suggest to use consequently 'winter' throughout the manuscript, as done in the caption of Fig. 3 just below.

'Years' will be switched to 'winters' when applicable.

5. Lines 113 & 116: Consider switching from '*accuracy*' to 'uncertainty'.

We will change 'accuracy' to 'uncertainty' in L.113 and L.116.

**Specific Comments:**

6. Throughout the text you use the term '*layer*' very loosely. While this term is clearly defined in the International Classification for Seasonal Snow on the Ground (ICSSG) [see also section 3.2.3, line 237] When reading '*layer*' here, it is often difficult to know what is meant. For example, at FOREST, what is the basal layer? On the other hand, you sometime use something like '*X % of the snowpack*' (see for example line 303). I would thus suggest to speak only in terms of such fractions or to visualize them in one profile. One particular example is found on lines: '… *with an evident decoupling between air and top layer temperatures beginning in early February.*' Here I understand you speak of the sensor located at a height of 64 cm, which is in the middle of the snowpack at that time (main reason for the observed decoupling) and not in what I would call '*the top layer*'.

Yes, the meaning of layer used in the manuscript does not always correspond to the definition in the ICSSG. We will change the word 'layer' when it is not referring to a layer according to the definition of the ICSSG.

7. The very useful concept of depth normalization needs to be discussed in more details. Indeed, that concept can be questioned in view of the potentially marked difference in total mass between '*normalised*' profiles, even taken at the same site. See also my comment on lines 93-94 below.

When looking at the total mass of the snowpack, the use of normalized profiles can certainly be questioned. However, in this study we use the normalization to compare the vertical variations of density and thermal conductivity between different snow pits. We do not look at the total mass of the snow in each pit. For this reason, we think that in our case the use of normalized profiles is appropriate.

8. Section 3.3.1, lines 276-291: I found this section difficult to read and not free of ambiguities. I would suggest to reshape it and concentrate on the salient features due to the implementation of the three key processes. Furthermore, in the caption of Fig. 9 you state that a '*blowing snow module is implemented*' at FOREST. I think this should be better stated in the text, along with recalling what processes (Snowfall, Compaction, Blowing snow) are activated at both sites, referring to section 2.3 for details.

We will clearly state in section 3.3.1 what processes have been activated and we will try to make it clearer:

'At TUNDRA, the default version shows reasonable agreement with the observations. In winter 2018–19, snow height is underestimated by 15 to 20 cm during the accumulation period and the snow disappearance date is 12 days earlier than the observations. For winter 2019–20, there is better agreement between the default version of Crocus and the observations, with a mean negative bias of 10 cm, leading to a modeled melt-out date that is just two days later than the observed date. Simulations with the adjusted version of Crocus (including the processes *Compaction* and *Snowfall*) for TUNDRA show an increased snow height of 10 to 20 cm compared to the default version. For winter 2019–20, this leads to a delayed melt-out that is 15 days later than the observed date while it is closer to observations in winter 2018-19. One striking difference between the two versions is that the snow height fluctuations are dampened in the adjusted version of Crocus due to the reduced compaction and the heavier fresh snow.

Since the meteorological forcing is nearly the same at both sites, it is not surprising that the snow heights modeled by the default version of Crocus at FOREST are very similar to those at TUNDRA. As a result, the modeled snow heights of the default version are lower than the observed snow heights by a factor of 2. Thus, the simulated melt-out date is early by one month for winter 2018–19 and by 16 days for winter 2019–20. The adjusted version of Crocus (including the processes *Compaction*, *Snowfall* and *Blowing Snow*) simulated the snow heights that are much closer to those observed. However, snow heights are still underestimated by 10 cm to 70 cm. The melt-out date is better simulated, with a difference of 9 days in winter 2018–19 and 0 days in 2019–20.'

9. Line 15: '*to some extent*' to be deleted. Having read the discussion and the conclusions I feel the adjustments are site-specific indeed.

We agree that the adjustments likely are site-specific but to which degree needs to be investigated in future studies in order to determine if they are only applicable at this site or at other sites as well. The similarities between our adjustments and those introduced in Barrere et al. (2017), Gouttevin et al. (2018) and Royer et al. (2021) suggest that at least some of the concepts are indeed applicable to different sites. We will rephrase the sentence and replace 'but are site-specific to some extent' with 'but may not be applicable at other sites'

10. Lines 28-39: '*The weather conditions to which Arctic snow is typically exposed differ considerably from conditions in the boreal forest.*' Does '*Arctic snow*' refer to the tundra? I wonder whether the connection to your site in the forest-tundra ecotone could be better linked to the Umiujaq site where weather conditions are almost alike.

We were referring to the general characteristics of snow in the Arctic (north) vs. the boreal forest (further south), not necessarily at the boundary between the two as is the case in our study. To make it clearer, we rewrote this sentence: "The weather conditions to which Arctic snow is typically exposed differ considerably from conditions **generally found** in the boreal forest **further south**."

11. Lines 80-81: What was the vertical spacing between the needle probes? Was it identical at both sites? And what about the second pole at TUNDRA? Was the latter used in the analysis?

The heights of the needles (measured from the surface of the lichen) was 4, 14, 29, 44, 64 at TUNDRA and 4, 14, 29 and 64 cm at FOREST. We used the second pole at TUNDRA to compare its results with the other pole. As these data were not used in this study, we will delete the reference to the second pole. Instead, we will include the heights of the needles as follows:

"Snow temperature and thermal conductivity were measured at both sites using vertical poles equipped with TP02/TP08 heated needle probes (Hukseflux, The Netherlands)**. At TUNDRA, five needles were installed at heights of 14, 14, 29, 44, 64 cm above the ground, whereas four needles were installed at FOREST at heights of 4, 14, 29 and 64 cm.** The measurement principle[...]"

12. Lines 93-94: '*A 100 cm3 box cutter … and a field scale were used to measure the density profiles.*' At what spacing? Continuously all 3 cm? This is important to note in particular as I assume many more measurements were performed at FOREST than at TUNDRA.

The spacing varied between the snow pits, mostly it was measured continuously at a 3 cm spacing (also at FOREST) but sometimes the spacing was increased to 5 cm. We will mention this in the manuscript:

'**The spacing was mostly 3 cm between measurements but was increased to 5 cm for some snow pits, essentially those with thicker snow**.'

13. Lines 144 & 146: '*we selected a fixed value of 0.05*' Read that way, it sounds it was always applied. It would be very helpful to move up here the remark on lines 146-147.

We will move up L. 146-147 so that it directly follows the statement mentioned. Thanks.

14. Table 1: Why was the long wave component not included?

We did not include the longwave component as no significant differences were observed. In general, radiation should be rather similar between the two sites due to the short distance between them. We further investigated the shortwave part (and showed it in the manuscript), because we assumed some influence of the steep ridge close by in combination with the lower elevation of FOREST resulting in a higher topographic shading. For the longwave counterpart, this should not play a significant role and as the tree density is very low at FOREST, no significant difference in longwave radiation was expected.

We will add a short explanation of why we did not show longwave radiation:

'Due to the low density of trees, no significant differences in longwave radiation were observed'

15. '*difference*': Was it TUNDRA - FOREST or vice versa? It matters wrt the mean and you need to add a 'Δ' in the table header.

Thanks for the remark, it was TUNDRA-FOREST. We will include this in the caption of the table as well as the 'Δ' in the header.

16. Line 184: '*fixed threshold of 0.5 °C*' Based on observations?

As we detailed in L. 184-186, the threshold was obtained by comparing manual observations of precipitation type at the nearby airport with the measured air temperature at the site.

17. Line 188: '*specific humidity from TUNDRA*' Is this not questionable in that case. The value of specific humidity influences turbulent fluxes, which may be quite different at both sites.

We agree that the specific humidity has a significant influence on the turbulent fluxes. However, at FOREST there are also measurements of this variable available. Unfortunately, there is only a short period when the instrument worked properly. Thus, we decided not to use the measurements as forcing but instead used them to compare the relative humidity at both sites. We concluded that it was sufficiently similar in order to use the same relative humidity for both sites. See comment 19. in the response to reviewer 2.

18. Line 233: '*melt-freeze forms were often present within these basal layers*' What did trigger these melt-freeze events? Turbulent fluxes? Long wave radiation from the nearby trees? I think this could be of importance when discussing the different processes at work at both sites.

We will include our hypothesis in the paper:

'We hypothesize that the melt-freeze crystals are formed at both TUNDRA and FOREST due to short warm spells at the beginning of the winter. However, at TUNDRA, the high values of the temperature gradient in December and January (compare Figure 11) trigger a more intense recrystallization compared to the FOREST site which and the melt-freeze form disappear at TUNDRA while they remain at FOREST.

19. Line 240: '*similar environments*' Can you clarify how far from the TUNDRA site this may be? This could be included in the methodology section though.

Usually, the snow pits were dug within 500 m of the TUNDRA station. As we were asked by reviewer 1 (comment 18)  to include a table containing all the snow pits, we will add the geographic location.

20. Line 241: 'In *order to make the profiles comparable, the snow heights were normalized.*' While I agree this is neat, I question whether it is straightforward wrt to the variability of HS, from winter to winter at TUNDRA, within a winter at FOREST (see Fig. 4.)?

It certainly covers differences in snow height at the different pits and times. However, the idea of the normalized heights is that regions of the snowpack which are subject to the same influences are shown at the same normalized height. For example, the upper parts of the snowpacks are affected by strong winds,  incoming radiation and other external influences which are all independent of the snow height. So effectively, even when there are differences in snow height they do not impact the conclusions drawn from the normalized profiles.

21. for Forest, it is more questionable for Tundra as ΔHS may be up to 200%

We do not compare snow pits from the beginning of the winter with others from the end. Most of the snow pits were dug in March, so differences are not as high as 200%. Furthermore, the interannual

comparison of the snow heights at FOREST shows that there is a strong resemblance of the evolution of the snow height from one winter to another.

**22.** Line 247: '*The scatter …*'Could you also say something about how many profiles did or did not follow the trend of the mean?

It is hard to come up with a definition or criterion that defines when a certain profile follows the trend of the mean or when it does not. We will publish all data, including the snow profiles, so the reader can compare the mean with every single profile directly.

**23.** Figure 8: It looks like part of the time you show temperatures at either 53 cm (T) or 64 cm (F) the sensors where not covered by at least 10 cm of snow. Please clarify or adjust the figure.

First, please note that the snow height is not necessarily the same at the snow gauge and where the snow temperatures are measured. Nonetheless, at the beginning of the period where we show the temperatures at 64 cm, the sensors might not be covered with 10 cm of snow.  At FOREST, there is no time-lapse camera and at TUNDRA it is a several meters away from the post, so visual inspections of the snow height at the posts are either impossible (FOREST) or rather difficult (TUNDRA). Therefore, we mostly relied on temperature measurements themselves and decided to show the temperatures when a significant decoupling from air temperature took place. As measurements were taken every other day at 5 AM, heating through solar radiation is not an issue.

**24.** Lines 295-296: '*This mismatch between observations and simulation is due to the transport of snow by wind.*' It is not clear to me whether you refer to a modeled or an observed (how?) event. Please clarify.

Here we refer to observed transported snow and those events were determined using the time-lapse cameras. We will clarify this in the manuscript:

"This mismatch between observations and simulation is due **to observed events of snow transport** by wind."

**25.** Line 298: '*The mean observed density profiles …*' You need to explain how modeled profiles are averaged here. You do so in the caption only … and how is not at all clear to me. Please clarify.

For the mean modeled profiles, we considered the months of January to March in the years 2017 to 2020 and took one profile each week during these periods. The mean is then obtained by normalizing each profile and taking the mean. We will add a description of this procedure to the text:

'[...]' to the default and adjusted model runs in Figure 10. **For calculating the mean of the simulated profiles, one profile per week during the months January through March from 2017 to 2020 was taken, normalized and then the average profile was calculated.**

**26.** Line 318: '*The impact of air temperature differences on snow cover was modest, …*' What about the impact of turbulent fluxes? And what about incoming long wave?

There may arise differences in turbulent fluxes due to differences in wind speed, but as the turbulent fluxes are rather small, possible differences are even smaller and most likely negligible.

The same applies to the incoming longwave radiation, there are only very small differences between the sites. And given the close proximity of the FOREST site to the TUNDRA site, no large difference in long wave radiation are to be expected.

The only forcing variable that significantly differs is the wind speed and two distinct datasets were used for this variable for the two sites.

27. Line 330: '…, *snow is continuously transported from the upper parts* …' Confirmed by measurements of wind direction?

The prevailing wind direction is west, from the Hudson Bay. However, wind direction is still often such that it allows snow to be transported from the upper parts of the valley to its lower parts. Additionally, photos from the time-lapse cameras confirmed that snow is being transported in the direction towards the lower parts of the valley.

28. Line 341: '…, soil freezes earlier in the winter.' I may miss a point here, but from Fig. 4 the snow depth does not seem to be significantly different in early winter. Later on I agree. Could there be other reasons for this early and deeper freezing?

It is true, the difference is smaller but still it does make a difference for the freezing of the soil. However, there are also other reasons such as the higher soil water content at FOREST (see Figure 1 and 2)

[Figure]

Figure 1: Time series of the soil temperature and the soil water content at FOREST at depths 9 cm, 15 cm, 27 cm, 39 cm, and 50 cm for the year 2019.

Also, there is more organic litter of low thermal conductivity under taller vegetation (see Gagnon et al. 2019). This further slows down soil cooling.

[Figure]

Figure 2: Time series of the soil temperature and the soil water content at TUNDRA at depths 5 cm, 10 cm, 20 cm, 30 cm and 50 cm.for the year 2019.

As the different snow cover is not the only reason for the later freezing, we will change the wording in L. 341:

'**In part** due to the reduced snow height and fairly similar snow thermal conductivity, soil freezes earlier in the winter.'

29. Figure 11: Is the sign of the temperature gradient correct? I would expect the contrary assuming the vertical axis is taken positive upwards.

The sign of the gradient depends on the definition i.e. whether the temperature at the upper measurement height is subtracted from the temperature of the lower one or vice versa. Here, we subtract the temperature at the upper height from the temperature at the lower height. We will clarify this in the caption of the figure:

"Positive gradients indicate higher temperatures close to the ground surface than in higher snow layers."

30. Line 392: '… *the solution may be to actually include them in models.*' … which has been actually done here:

*Jafari, M., Gouttevin, I., Couttet, M., Wever, N., Michel, A., Sharma, V., Rossmann, L., Maass, N., Nicolaus, M., and Lehning, M.: The Impact of Diffusive Water Vapor Transport on Snow Profiles in Deep and Shallow Snow Covers and on Sea Ice, Front. Earth Sci., 8, 25 pp., https://doi.org/10.3389/feart.2020.00249, 2020.*

I am not saying it works in Arctic snow though.

Thank you for the remark, however, Jafari et al. (2020) only included diffusive water vapor transport while convective transport is believed to be the main process (Trabant and Benson, 1972). We will include this reference together with another one (Simson et al. 2021) and state that first attempts were already made to address this issue.

**References**

Barrere, M., Domine, F., Decharme, B., Morin, S., Vionnet, V., & Lafaysse, M.: Evaluating the performance of coupled snow–soil models in SURFEXv8 to simulate the permafrost thermal regime at a high Arctic site, *Geosci. Model Dev.,* 10, 3461–3479, https://doi.org/10.5194/gmd-10-3461-2017, 2017.

Gouttevin, I., Langer, M., Löwe, H., Boike, J., Proksch, M., & Schneebeli, M.: Observation and modelling of snow at a polygonal tundra permafrost site: spatial variability and thermal implications, *The Cryosphere,* 12, 3693–3717, https://doi.org/10.5194/tc-12-3693-2018, 2018.

Gagnon, M., Domine, F., and Boudreau, S.: The carbon sink due to shrub growth on Arctic tundra: a case study in a carbon-poor soil in eastern Canada, Environ. Res. Commun., 1, 091 001, https://doi.org/10.1088/2515-7620/ab3cdd, 2019.

Royer, A., Picard, G., Vargel, C., Langlois, A., Gouttevin, I., & Dumont, M. (2021) Improved simulation of Arctic circumpolar land area snow properties and soil temperatures. *Front. Earth Sci.* 9:685140. doi: 10.3389/feart.2021.685140, 2021.

Simson, A., Löwe, H., and Kowalski, J.: Elements of future snowpack modeling – Part 2: A modular and extendable Eulerian–Lagrangian numerical scheme for coupled transport, phase changes and settling processes, *The Cryosphere,* 15, 5423–5445, doi: 10.5194/tc-15-5423-2021, 2021.

Trabant, D. and Benson, C. S.: Field experiments on the development of depth hoar, Geol. Soc. Am. Mem., 135, 309-322, 1972.

---

## Referee Report (RR1)

The authors well updated the manuscript. The main subject of this study is to show that water vapor transport is an important process shaping the vertical snow density profile in both tundra and forest-dominated areas and to test the performance of the snow model Crocus and explore the adjustments, that is clearly stated in the main text. I understood that the scientific originality is to present the snowpack structure and thermal properties in **both tundra and boreal forest** which no previous study addressed yet. Following these updates, I tried to check the manuscript focusing on the points strongly related to the main subject. However, some concerns that were pointed out in the previous round of review were not significantly solved.

First, the logical flow to conclude "predominance of water vapor fluxes for vertical density/thermal conductivity profile" is still insignificant. According to L422–425, we guess the authors reached the conclusion based on two results: (1) a significant fraction of depth hoar which is a remarkable characteristic indicating active water vapor flux, and (2) the snow density profile increasing towards the snow surface which should be decreasing towards the surface based on the compression process only. However, the strong wind densifies the snow surface, which coincides with high thermal conductivity (*wind packing*). The density of precipitation particles under the strong wind itself is also high (e.g. $\rho_n$=418 kg m-2 with $T_a$=263.15 K and $W_s$=4 m/s using Eq. 2 above the vegetation height, which is comparable to the observation). These natural processes and the stabilizing effect of shrubs can also cause the vertical density profile which is different from the expected one based on the compaction process only (the result by the default Crocus). The "predominance" should be argued after the order of importance in things is clarified. The authors should address each dependency of the process (compression, water vapor transport, stabilizing effect, and blowing snow effect) on the density profile quantitatively. Without this kind of quantitative discussion, it is difficult for me to understand that the water vapor transport **overcomes** the effect of the compression process.

Second, the robustness of density/thermal conductivity profiles observed is not enough demonstrated. The problem that the thermal conductivity observation using TP02/TP08 heated needle probes gave a large error was well discussed in the revised manuscript (though a small mistake was found; see a specific comment). I understand that the error would not be a problem in terms of average. However, the profiles were largely variated (Figs. 6 and 7). Whereas it seems that one profile increased towards the surface while another decreased, it is very difficult to see the true variability from a spaghetti plot. The authors should show the confidence interval (or probability density function) or perform a statistic test for the vertical tendency of the profile. Then, the robustness of the vertical tendency of the profile should be discussed. Even though the authors showed comparable profiles between the mean and median in "tc-2022-19-author_response-version2.pdf", that is only a piece of evidence that the probability density function might be assumed as the normal distribution.

Third, much evidence for coefficients of the adjustments (Eq. 1–4) is not still disclosed even though exploring adjustments is one of the study subjects. This is an important point in terms of scientific reproducibility, too. The authors should appropriately show the methods and results of preliminary simulations (L137–138, L152–153, L167–168) and discuss the improvements from the original adjustments by Barrere et al. (2017), Gouttevin et al. (2018), and Royer et al. (2021b). Although the authors were concerned as "Including an analysis would give the impression that we recommend the used parameters for other sites and/or models", that is not a problem since the authors appropriately made remarks about it at L413–420. Moreover, I guess that the coefficients $a\_\rho$, $b\_\rho$, $c\_\rho$, a, b, and c are intrinsically obtained based on the multivariable model with a very large degree of freedom: the density and snow height are outcome variables and $a\_\rho$, $b\_\rho$, $c\_\rho$, a, b, and c are explanatory variables. Eq. (1) also affects the selection of coefficients. Since this seems very complex model to obtain the coefficients, the authors need to describe the procedure carefully.

Specific comments:

1. L46–48: This sentence should be revised as "These models have been developed for alpine applications, where the dominant process that controls the density profile in the models is the compaction that results from overburden."

2. L48–50: This parameterization is ad-hoc to fit the calculation result with the observation as the authors say in "tc-2022-19-author_response-version2.pdf". The water vapor transport has not been directly implemented. Please emphasize this point here and revise this sentence appropriately.

3. Fig. 1: Please describe what a dotted line of the figure indicates in the caption.

4. L108–116: First, you indicate the blowing snow effects consist of snow erosion and accumulation (L109). However, after that, you say sublimation and wind packing effect are implemented in Crocus to simulate blowing snow (L110, L115). This is very confusing because the sublimation and wind packing effects are strictly different from erosion and accumulation, respectively. Please revise the wordings appropriately.

5. L128–143: The terminology of *Snowfall* is very confusing. According to L141–143, *Snowfall* includes densification effects of the wind, which associate with *Blowing snow*. Please replace it with *Snowfall density* or something appropriate word. Moreover, the densification effects of the wind have been already considered in Eq. (1) as a wind packing effect. Is this an appropriate procedure? Please clarify it in the main text.

6. L137–138: It is very hard to understand how the authors obtain $a\_\rho$ by varying $c\_\rho$ to be comparable density between the simulation and observation. Please describe the procedure

more!

7. L156: The terminology of *Blowing snow* is confusing because this includes snow erosion, accumulation, and wind packing until this sentence. Please replace it with *Snow redistribution* or something appropriate words.

8. L163: Please delete "or remove" from this sentence. This is confusing because this study does not remove snow using Eq. (4).

9. Eq. (4): You can replace "W_s" with "min(W_s, 10)".

10. L173–174: Please delete this sentence. This is confusing because this study does not remove snow using Eq. (4).

11. L182: Please replace "ERA5" with "the closest grid point data of ERA5 to the study site".

12. L185: Please replace "rain and snow" with "liquid and solid".

13. L188: Typo.

14. L236: Why do you need to simplify stratigraphy? Fig. 5 and Supplementary figure 2 are very similar.

15. L244: However, melt-freeze forms are depicted at the bottom in Supplementary figure2. Please make sure if there is no mistake.

16. L246: However, no precipitation particles are depicted in Supplementary figure 2. Please make sure if there is no mistake.

17. L252: Please revise as "…, every measurement height was divided by the height of the respective snow cover."

18. L254: However, the variability among the measurements is very large. Is the profile really different between TUNDRA and FOREST significantly?

19. L327: I feel like there is something a little bit off with this sentence because the adjustments were ad-hoc procedures to fit the calculation result with the observation. Is it appropriate to say "the adjusted version **does reproduce** the density profile …"? Please reconsider the terminology.

20. L351: Snow amount at TUNDRA would be expected to be reduced by a strong wind. However, Crocus simulation, which ignores the snow erosion, relatively simulates snow height (Fig. 9) and snow water equivalent (Supplementary figure 1). How do you interpret this result? Please describe it in the main text.

21. L354–357: Do you have some cases when the depth hoar was not or weakly developed with much snow amount? In such cases, if this hypothesis was true, the difference between the default Crocus simulation and the observation should be smaller.

22. L380: Does "the uncertainty" indicate the standard error? If so, "n-1" should be correctly "n-1/2". Please make sure if there is no mistake.

23. L403–404: So, did your implementation, not taking the whole vegetation height as a zone

where compaction is reduced, effectively improve the simulation score? This can easily demonstrate quantitively through the preliminary sensitivity analysis (L153). Even though you responded to my comment as "we did not focus on a quantitative assessment…" in "tc-2022-19-author_response-version2.pdf", the quantitative assessment is necessary to demonstrate the **predominance** of water vapor transport.

24. Supplementary figure1: This should move to the main text.
25. Supplementary figure2: No legend for "/".

---

## Author Response (AR2)

**Responses to Reviewer #1**

**General Comments:**

The authors well updated the manuscript. The main subject of this study is to show that water vapor transport is an important process shaping the vertical snow density profile in both tundra and forest-dominated areas and to test the performance of the snow model Crocus and explore the adjustments, that is clearly stated in the main text. I understood that the scientific originality is to present the snowpack structure and thermal properties in both tundra and boreal forest which no previous study addressed yet. Following these updates, I tried to check the manuscript focusing on the points strongly related to the main subject. However, some concerns that were pointed out in the previous round of review were not significantly solved.

We did our best to resolve remaining issues.

1.First, the logical flow to conclude "predominance of water vapor fluxes for vertical density/thermal conductivity profile" is still insignificant. According to L422–425, we guess the authors reached the conclusion based on two results: (1) a significant fraction of depth hoar which is a remarkable characteristic indicating active water vapor flux, and (2) the snow density profile increasing towards the snow surface which should be decreasing towards the surface based on the compression process only. However, the strong wind densifies the snow surface, which coincides with high thermal conductivity (wind packing). The density of precipitation particles under the strong wind itself is also high (e.g. $\rho\_n$=418 kg m-2 with $T\_a$=263.15 K and $W\_s$=4 m/s using Eq. 2 above the vegetation height, which is comparable to the observation). These natural processes and the stabilizing effect of shrubs can also cause the vertical density profile which is different from the expected one based on the compaction process only (the result by the default Crocus). The "predominance" should be argued after the order of importance in things is clarified. The authors should address each dependency of the process (compression, water vapor transport, stabilizing effect, and blowing snow effect) on the density profile quantitatively. Without this kind of quantitative discussion, it is difficult for me to understand that the water vapor transport overcomes the effect of the compression process.

First of all, the presence of depth hoar is a non-ambiguous demonstration of the existence of large water vapor fluxes. Numerous studies such as (Trabant and Benson (1972), Strum and Benson (1997), Domine et al. (2016)) have demonstrated that in the Arctic, the water vapor fluxes are large enough to trigger a vertical redistribution of the snow mass. Second, we do not think that wind compaction can be invoked to affect significantly the density profile. First, wind compaction affects all layers, not just surface layers, as seen during our field trips, usually in late March. Second, our climatology monitoring at TUNDRA site and FOREST has shown that wind speed is greater in fall compared to mid-winter, so

that it affects basal snow layer more strongly. Top layers are therefore in general less likely to be strongly compacted by wind than lower layers. It sure can be suggested that shrubs would contribute to a lower basal snow density. However, our observations show that birch shrubs are very strongly compacted. Time lapse images show that this happens very early in the season so that their impact is very limited, and is in fact probably visible only in the lowest 10 to 15 cm of our density profiles (Figure 6). In fact, it is well known that birch stems are very supple and rapidly get compressed by snow, which limits their effect on the snow density profile. Sturm et al. (2005) mention (their page 6) "The 1.5- to 3-cm-diameter birch stems (Betula glandulosa) at the tall shrub site were more supple than the 3- to 5-cm-diameter willow (Salix pulchra) stems at the woodland site." These authors also observed that they were quickly compacted. Finally, our modeling work shows that simulations without water vapor fluxes clearly cannot reproduce observations. This is a strong indication that water vapor fluxes is the missing process in models. Our attempt to improve simulations by adjusting the density parameterization is more an error compensation scheme, a common strategy in modeling, but this cannot be used to negate the crucial role of water vapor fluxes in determining the observed density profiles.

2.Second, the robustness of density/thermal conductivity profiles observed is not enough demonstrated. The problem that the thermal conductivity observation using TP02/TP08 heated needle probes gave a large error was well discussed in the revised manuscript (though a small mistake was found; see a specific comment). I understand that the error would not be a problem in terms of average. However, the profiles were largely variated (Figs. 6 and 7). Whereas it seems that one profile increased towards the surface while another decreased, it is very difficult to see the true variability from a spaghetti plot. The authors should show the confidence interval (or probability density function) or perform a statistic test for the vertical tendency of the profile. Then, the robustness of the vertical tendency of the profile should be discussed. Even though the authors showed comparable profiles between the mean and median in "tc-2022-19- author_response-version2.pdf", that is only a piece of evidence that the probability density function might be assumed as the normal distribution.

The reviewer invokes the spatial and temporal variability of our observations to minimize the strength of our arguments. We feel this is not reasonable. Any field snow scientist knows that snowpack vertical profiles are highly variables in both time and space and that apprehending the properties of snowpacks at a given location can only be done by averaging, as we have done. These averages do show significant trends and with reason based on those.
Several options to illustrate the variability of the profiles exist. We prefer to keep the use of spaghetti plots because it allows to show the range of the observed values, while the single profiles are still visible. As suggested by the reviewer, we included the standard deviation of each normalized height increment in the figures containing the profiles.

[Figure]

Figure 6: Snow density profiles from 29 snow pits near TUNDRA and 18 snow pits near FOREST collected between January and March from the years 2012 to 2019. For better comparability, snow heights were normalized. The means of all profiles are also shown, together with the standard deviation.

However, as mentioned before the median is a statistically robust measure and therefore guarantees the robustness of the vertical profiles.

3. Third, much evidence for coefficients of the adjustments (Eq. 1–4) is not still disclosed even though exploring adjustments is one of the study subjects. This is an important point in terms of scientific reproducibility, too. The authors should appropriately show the methods and results of preliminary simulations (L137–138, L152–153, L167–168) and discuss the improvements from the original adjustments by Barrere et al. (2017), Gouttevin et al. (2018), and Royer et al. (2021b). Although the authors were concerned as "Including an analysis would give the impression that we recommend the used parameters for other sites and/or models", that is not a problem since the authors appropriately made remarks about it at L413–420. Moreover, I guess that the coefficients $a_\rho$, $b_\rho$, $c_\rho$, a, b, and c are intrinsically obtained based on the multivariable model with a very large degree of freedom: the density and snow height are outcome variables and $a_\rho$, $b_\rho$, $c_\rho$, a, b, and c are explanatory variables. Eq. (1) also affects the selection of coefficients. Since this seems very complex model to obtain the coefficients, the authors need to describe the procedure carefully.

As stated in the manuscript, we did not conduct any multivariable model analysis to obtain the parameters. Instead, we prescribed them ourselves by visually comparing simulated and observed profiles. This is one reason why we specifically emphasize that we do not recommend the use of these specific parameters and equations in other studies. As for scientific reproducibility, it is ensured by presenting all the parameters used, in addition to the fact that the default and adjusted versions of the model are available online.

As for the other studies, directly comparing them is not possible as all used different sites and a different model. For instance, SNOWPACK was used in Gouttevin et al. (2018) in Siberia. Barrere (2017) modeled a site in the High Arctic almost 2000 km further north, while Royer et al. (2021) made Arctic-wide simulations. All these studies obtained site-specific fitting parameters that were all error compensation tricks to make up for the lack of description of water vapor fluxes. We agree that a detailed analysis comparing all the parametrizations at all the sites could be useful to investigate the capabilities of such simple schemes to improve simulations of the snowpack in the Arctic.

**Specific comments:**

4. L46–48: This sentence should be revised as "These models have been developed for alpine applications, where the dominant process that controls the density profile in the models is the compaction that results from overburden."

We thank the reviewer for this suggestion. However, the sentence rephrased in this way suggests that the dominant process controlling the density profile is overburden only in the models, whereas this process is dominant in both nature and the models. We therefore prefer to keep the current wording.

5. L48–50: This parameterization is ad-hoc to fit the calculation result with the observation as the authors say in "tc-2022-19-author_response-version2.pdf". The water vapor transport has not been directly implemented. Please emphasize this point here and revise this sentence appropriately.

We inserted a statement to emphasize that no water vapor transport was simulated:

To overcome the lack of water vapor transport, Barrere et al. (2017), Gouttevin et al. (2018) and 'Royer et al. (2021b) all introduced modifications, **without explicitly simulating water vapor transport**, by increasing the maximum density of wind-induced snow compaction and adapting compaction to vegetation characteristics.'

6. Fig. 1: Please describe what a dotted line of the figure indicates in the caption.

A description of the dotted line was integrated into the caption of Figure 1, thank you. We mentioned that the dotted line is the boundary between the forested area and the shrub and lichen tundra.

7. L108–116: First, you indicate the blowing snow effects consist of snow erosion and accumulation (L109). However, after that, you say sublimation and wind packing effect are implemented in Crocus to simulate blowing snow (L110, L115). This is very confusing because the sublimation and wind packing effects are strictly different from erosion and accumulation, respectively. Please revise the wordings appropriately.

We revised L.109 to mention that additional effects come along with snow erosion and and accumulation.

'However, the 1D nature of the model does not allow a direct simulation of snow erosion and **accumulation and the associated effects (additional compaction and increased sublimation rates).**'

8. L128–143: The terminology of Snowfall is very confusing. According to L141–143, Snowfall includes densification effects of the wind, which associate with Blowing snow. Please replace it with Snowfall density or something appropriate word. Moreover, the densification effects of the wind have been already considered in Eq. (1) as a wind packing effect. Is this an appropriate procedure? Please clarify it in the main text.

The terminology *Snowfall* sums up all effects related to the deposition of snow and the snowpack surface. We prefer the use of a single word for the sake of brevity. The density effects described in Eq. 1 were not sufficient to account for the observed densification. For this reason, we included the process Snowfall.

9. L137–138: It is very hard to understand how the authors obtain a_ρ by varying c_ρ to be comparable density between the simulation and observation. Please describe the procedure more!

Sorry for the confusion. In fact, we vary both to obtain a comparable density profile in simulations as observed on site, as detailed in the manuscript. It should be noted that we had omitted a_ρ in L. 137-138, which probably contributed to the confusion. We rephrased as follows:

"These values were obtained with a sensitivity analysis where we varied $a_\rho$ **and** $c_\rho$ in order to obtain a good agreement between the simulated and observed densities of the top of the snow cover."

10. L156: The terminology of Blowing snow is confusing because this includes snow erosion, accumulation, and wind packing until this sentence. Please replace it with Snow redistribution or something appropriate words.

The process *Blowing Snow* was included to account for the snow accumulation at FOREST. We fear that using *Snow redistribution* would give the impression that we actually redistribute the snow from one point to another (e.g. TUNDRA to FOREST), which is not the case.

11. L163: Please delete "or remove" from this sentence. This is confusing because this study does not remove snow using Eq. (4).

We removed the 'or remove' from the manuscript.

12. Eq. (4): You can replace "W_s" with "min(W_s, 10)".

Thank you for your suggestion, we presented the equation using min(U,10) as suggested (U replaced Ws as requested by reviewer Dr. Charles Fierz).

13. L173–174: Please delete this sentence. This is confusing because this study does not remove snow using Eq. (4).

The sentence was removed.

14. L182: Please replace "ERA5" with "the closest grid point data of ERA5 to the study site".

ERA5 was replaced with the suggestion given.

15. L185: Please replace "rain and snow" with "liquid and solid".

Thank you, we followed your suggestion.

16. L188: Typo.

Thanks for pointing out the typo, we corrected it.

17. L236: Why do you need to simplify stratigraphy? Fig. 5 and Supplementary figure 2 are very similar.

We simplified it to emphasize the important characteristics.

18. L244: However, melt-freeze forms are depicted at the bottom in Supplementary figure 2. Please make sure if there is no mistake.

Indeed, sometimes melt-freeze forms are present at TUNDRA, such as in this snow pit. As mentioned in the comment above, we simplified the profiles to emphasize the important and dominant characteristics of each site. Obviously, there is no real snow pit that shows exclusively these characteristics. In this specific snow pit, the melt-freeze forms were less abundant and less striking compared to FOREST. As such, we have chosen to omit them in the simplified profiles to present the typical profiles found at the site.

19. L246: However, no precipitation particles are depicted in Supplementary figure 2. Please make sure if there is no mistake.

The sign '/' in supplementary figure 2 stands for 'Partly decomposed precipitation particles'. Thus, these are slightly broken precipitated particles. As for the comment before, we made the choice to alter the profiles slightly to highlight the important characteristics.

20. L252: Please revise as "..., every measurement height was divided by the height of the respective snow cover."

The sentence was revised as suggested.

21. L254: However, the variability among the measurements is very large. Is the profile really different between TUNDRA and FOREST significantly?

Please see answer to comment 2.

22. L327: I feel like there is something a little bit off with this sentence because the adjustments were ad-hoc procedures to fit the calculation result with the observation. Is it appropriate to say "the adjusted version does reproduce the density profile ..."? Please reconsider the terminology.

We did not adjust the model to the mean of the profiles but to single profiles. It is thus not self-evident that the mean of the model reproduces the mean of the observations. Particularly given that the mean in the model contains many profiles from January through March over several years.

23. L351: Snow amount at TUNDRA would be expected to be reduced by a strong wind. However, Crocus simulation, which ignores the snow erosion, relatively simulates snow height (Fig. 9) and snow water equivalent (Supplementary figure 1). How do you interpret this result? Please describe it in the main text.

As already stated in the text, the snow height further up the valley was found to be very small. Thus, we concluded that snow at TUNDRA does get eroded but there is also deposition of snow coming from further up in the valley. We added this to the manuscript:

'**As snow erosion and deposition at TUNDRA likely balance each other,** the snow height at TUNDRA is more closely correlated with the precipitation rate, which is typically low from January to March (see Figure 3).'

24. L354–357: Do you have some cases when the depth hoar was not or weakly developed with much snow amount? In such cases, if this hypothesis was true, the difference between the default Crocus simulation and the observation should be smaller.

Depth hoar or indurated depth hoar was found in all snow pits.

25. L380: Does "the uncertainty" indicate the standard error? If so, "n-1" should be correctly "n-1/2". Please make sure if there is no mistake.

Yes, this is a typo. It should be $n^{-1/2}$. Thank you for spotting this.

26. L403–404: So, did your implementation, not taking the whole vegetation height as a zone where compaction is reduced, effectively improve the simulation score? This can easily demonstrate quantitively through the preliminary sensitivity analysis (L153). Even though you responded to my comment as "we did not focus on a quantitative assessment..." in "tc-2022-19-author_response-version2.pdf", the quantitative assessment is necessary to demonstrate the predominance of water vapor transport.

The choice to include only part of the vegetation height slightly worsened the results. However, this choice was made as we observed bent shrubs in the snow pits and as we demonstrate in the manuscript, the bending of shrubs has already been studied (Ménard et al. (2014) and Belke-Brea et al. (2020). Thus, we feel it is a scientifically sound approach to take only part of the vegetation height to stabilize the snow.

27. Supplementary figure1: This should move to the main text.

We agree that this figure might be more interesting for people interested in snow hydrology, but in this study, we focus on the density profiles and not the SWE. Thus, our primary choice would be to leave it in the supplementary material.

28. Supplementary figure2: No legend for "/".

Thank you for the remark. This legend was added.

**Responses to Dr. Charles Fierz**

**General comments**

I carefully went through all replies of the authors and the changes to the original submission. The authors adequately responded to the issues of the reviewers and in particular those I addressed. So I find the manuscript to have substantially improved. It is know also clearer that there is no direct evidence of the importance of water vapor transport from simulations as there is no such process implemented in the model. A few of my suggestions below could help further disentangle this aspect that was strongly pinpointed by reviewer #1. Of course, with major revisions, a few new questions arise that are reflected in my comments below.

In summary I recommend accepting the paper after the authors addressed these minor issues.

Thanks again for your comments. We also believe that they have helped us to improve the quality of the manuscript.

**Comments to your replies**

1. Point 2: Your choice, no problems. But the reason for it is not overwhelming. Snow depth – or here snow height – is defined as "… the total height of the snowpack, i.e., the vertical distance in centimetres from base to snow surface." Or even more inclusive the "vertical distance from the snow surface to a stated reference level." It is thus not associated with the direction of the vertical axis.

We agree that the difference is minor between snow depth and snow height.

2. Point 20: I eventually agree with you, in particular because all your snow pits were recorded around the same time in the year.

Indeed, comparing snow profiles from the very beginning of the winter with ones from the end of the winter could be more problematic and misleading.

3. Point 29: 'The sign of the gradient depends on the definition' I'd rather say it depends on the sign of the vertical axis that you obviously define as positive upward. Accordingly, what you call 'positive' is mathematically negative. Please correct.

Yes, we meant that the sign of the gradient depends on the definition of the direction of the vertical axis. This was corrected.

4. Point 30: '… while convective transport is believed to be the main process.' Thus the question is not settled for now! While I agree convection may be difficult to include in 1D models, Jafari et al. (2022) presented a numerical 2D study that may help setting the conditions for convection to occur in snowpacks. See Jafari, M., Sharma, V., and Lehning, M.: Convection of water vapour in snowpacks, 934, https://doi.org/10.1017/jfm.2021.1146, 2022.

The question might not be completely settled, however, Jafari et al. (2022) state themselves that 'It has been concluded that significant convection must occur in snowpacks to explain the observations [...] and diffusion of water vapor is too slow to explain observations (Domine et al. (2016), Fourteau et al. (2021)). Thus, it appears highly unlikely that diffusion is the main process responsible for the vertical water vapor transport. We fully agree that studying 2D convection can help to understand conditions when it occurs and that 1D models might benefit from those findings.

**Specific comments to the revised version**

5. Lines 12-13: I suggest to move the sentence starting, "_We demonstrate …_" to Line 10 after the sentence ending, "… typical of the Arctic." such that it becomes clearer that no water vapor transport was implemented in Crocus.

Thanks for the suggestion, we swapped these two sentences.

6. Line 145: I am afraid your notation for wind speed W s is still inadequate. Variables are usually noted with one letter only, thus my proposal to use U instead. If you still want to use two letters, use WS (in roman typeface and not italic). Apply change in Figure 2 too.

We changed Ws to U.

7. Lines 252-253: '…, the snow heights were normalized' In fact, the snow depth - or snow height here is the normalization factor. Consider changing to "…, the heights were normalized by the snow height.". The same applies to the captions of Figures 6 & 7.

Good remark, we changed this.

8. Lines 308-312: I am asking myself whether the 'discrepancy' is not rather due to mismatch in depth of snowfall? At least your comment on the situation at FOREST points towards it. The wording may have to be adjusted accordingly.

We do not have enough measurements of SWE to track its evolution over the course of a winter. Thus, there is some uncertainty when it comes to a comparison between the observed and modeled SWE. However, the large overestimation of the SWE at FOREST suggests that there is a problem with the total mass of the snow, rather than just in the depth of the snowfall.

9. Lines 454-456: Like in the abstract, I think the sentence starting, "However, …" is misplaced here. It should be moved after the sentence ending on Line 451 and may need some adjustment in wording.

We moved the sentence to the indicated position.

10. Figure S1: A pity you do only have one profile at FOREST in 2019! An additional point would have been interesting, particularly to show the spread among measured profiles at FOREST (see comment to Lines 308-312). However, you may consider comparing the measured SWE to accumulated precipitation (see Figure 3) as they should somehow match, in particular during precipitation events. Of course, blowing snow, sublimation, and imperfect correction for undercatch add to the uncertainty may make it a cumbersome task …

Having more profiles is certainty always beneficial, however, considering the large snow height at FOREST, it was usually not feasible to dig several snow pits there. We agree that comparing the SWE to cumulative precipitation a priori might be interesting, however, considering the important snow transport by the wind, the cumulative precipitation and the SWE do not necessarily match (SWE in 2019 at FOREST: 535 mm; cumulative precipitation: 233 mm) and thus, no conclusions can be drawn from the comparison. To study the impact of wind and the topography, specialized measurements at the scale of the valley would be needed.

**Minor comments to the revised version**

All the minor comments were implemented in the manuscript except for comments 18 and 36 (see below for details).

11. Line 14: Replace 'The adjustments that were made to Crocus …' to "These adjustments …"
12. Line 156: Replace 'the lacking consideration of a' to "not considering any"
13. Line 166: If Pnew and Pold are rates, the unit should be kg m-2 s-1 and not just kg m-2.
14. Lines 186 &188: Replace '0.5° C' and '0.8° CC' to "0.5 °C" and "0.8 °C", respectively. Check throughout the text that there is a space between the value and the unit, that is x °C.
15. Lines 195-196: Replace 'of 0.4 m height (TUNDRA) and 1.3 m height (FOREST)' to "of 0.4 and 1.3 m height at TUNDRA and FOREST"
16. Line 199: Add 'However, as some …'
17. Fig. 2 and Table 1: 'SWdown' As for wind speed, use a one letter variable or use roman typefaces if you want to stick to SW. Also, note that descriptive subscripts are always in roman typeface . This is valid for all variables throughout the text (see also https://physics.nist.gov/cuu/pdf/typefaces.pdf)
18. Figure 3: Consider using "kg m-2" here too instead of 'mm'.

We prefer to stick to mm to make it comparable to indications of precipitation.

19. Figures 4, 9: Change y-axis label to "Snow height (m)"
20. Figures 5 & S2: The dates of the profiles have to match those in Tables S1 & S2 and given consistently in both figures. For example, I do not see any profile taken on 24 Feb at FOREST, but on 26th or 28th, the latter with Hs = 148 cm ?

Consider adding the snow depth to the dates, for example "(25 February, snow height is 69 cm)".

Indeed, the data was wrong and was corrected. Also in the drawing the snow height appeared to high to represent a height of 148 cm. This was corrected as well.

21. Figures 5 & S2: Change y-axis label to "Height (m)"
22. Figures 6, 7, 10: Change y-axis label to "Normalized height (m)"

The normalized height is no longer in the unit m. So we changed the label to 'Normalized height'.

23. Line 256: Replace 'snow layer' by "part"
24. Line 257: Replace 'this general' by "that decreasing"
25. Figure 8: Add missing y-axis label "Temperature"
26. Line 289: Replace 'disappearance' by "melt-out"
27. Line 308: Replace ' e.g. the total snow mass.' by ", i.e. the total snow mass per unit area."
28. Lines 356-357: Reword 'creating a greater vertical gradient' as "a large vertical gradient"
29. Figure 11: Add missing y-axis label "Temperature gradient"
30. Line 371: Reword 'not included relationships' as "often not included in relationships"
31. Line 378: Replace 'Generally' by "Admittedly"
32. Line 350: Add the geographical direction, I guess "~500 m north from TUNDRA"
33. Line 380: Replace 'the all' by "all"
34. Line 444: Replace 'wind-induced' by "wind-driven"
35. Figure S2: What do the thick lines in the FOREST profile indicate?

These are ice layers. We clarified this in the caption.

36. Tables S1 & S2: It may be interesting and valuable to indicate the measured bulk snow density (or/ and SWE) for all measured profiles in these tables, whenever possible.

Oftentimes, the resolution of the measurements does not allow the calculation of the bulk density. For this reason, and also because our focus lies purely on the density profile, we did not include a column with the bulk density. For those interested, the dataset containing all density values will be available together with a paper describing the dataset. These data were submitted to PANGAEA together with a full dataset of  soil and meteorological observations at Umiujaq TUNDRA and FOREST sites. Hopefully, the data set will be available if and when the paper is accepted and this will be indicated in the paper, so all the data will be available to readers.

37. Tables S1 & S2: Use the ISO-format yyyy-mm-dd for the dates.

**References**

Belke-Brea, M., Domine, F., Boudreau, S., Picard, G., Barrere, M., Arnaud, L., and Paradis, M.: New allometric equations for Arctic shrubs and their application for calculating the albedo of surfaces with snow and protruding branches, Journal of Hydrometeorology, 21, 1–49, https://doi.org/10.1175/JHM-D-20-0012.1, 2020.

Domine, F., Barrere, M., and Sarrazin, D.: Seasonal evolution of the effective thermal conductivity of the snow and the soil in high Arctic herb tundra at Bylot Island, Canada, The Cryosphere, 10, 2573–2588, https://doi.org/10.5194/tc-10-2573-2016, 2016.

Gouttevin, I., Langer, M., Löwe, H., Boike, J., Proksch, M., and Schneebeli, M.: Observation and modelling of snow at a polygonal tundra permafrost site: spatial variability and thermal implications, The Cryosphere, 12, 3693–3717, https://doi.org/10.5194/tc-12-3693-2018, 495, 2018.

Royer, A., Picard, G., Vargel, C., Langlois, A., Gouttevin, I., and Dumont, M.: Improved Simulation of Arctic Circumpolar Land Area Snow Properties and Soil Temperatures, Frontiers in Earth Science, 9, 515, https://doi.org/10.3389/feart.2021.685140, 2021b.

Ménard, C. B., Essery, R., Pomeroy, J., Marsh, P., and Clark, D. B.: A shrub bending model to calculate the albedo of shrub-tundra, Hydrological Processes, 28, 341–351, https://doi.org/10.1002/hyp.9582, 2014.

Sturm, M. and Benson, C. S.: Vapor transport, grain growth and depth-hoar development in the subarctic snow, Journal of Glaciology, 43, 42–59, https://doi.org/10.3189/S0022143000002793, 1997.

Sturm, M., and Benson, C. S.: Vapor transport, grain growth and depth-hoar development in the subarctic snow, J. Glaciol., 43, 42-59, 10.3189/S0022143000002793, 1997.

Sturm, M., Douglas, T., Racine, C., and Liston, G. E.: Changing snow and shrub conditions affect albedo with global implications, Journal of Geophysical Research-Biogeosciences, 110, G01004, 10.1029/2005jg000013, 2005.

Trabant, D., and Benson, C. S.: Field experiments on the development of depth hoar, Geol. Soc. Am. Mem., 135, 309-322, https://doi.org/10.1130/MEM135-p309, 1972.